# Frozen Policy Iteration: Computationally Efficient RL under Linear $Q^\pi$ Realizability for Deterministic Dynamics

**Yijing Ke**[1], **Zihan Zhang**[2], **Ruosong Wang**[3]

[1]School of EECS, Peking University
[2]Department of Computer Science and Engineering, HKUST
[3]CFCS and School of Computer Science, Peking University
`keyijing@stu.pku.edu.cn`
`zihanz@cse.ust.hk`
`ruosongwang@pku.edu.cn`

## Abstract

We study computationally and statistically efficient reinforcement learning under the linear $Q^\pi$ realizability assumption, where any policy's $Q$-function is linear in a given state-action feature representation. Prior methods in this setting are either computationally intractable, or require (local) access to a simulator. In this paper, we propose a computationally efficient online RL algorithm, named *Frozen Policy Iteration*, under the linear $Q^\pi$ realizability setting that works for Markov Decision Processes (MDPs) with stochastic initial states, stochastic rewards and deterministic transitions. Our algorithm achieves a regret bound of $\widetilde{O}(\sqrt{d^2 H^6 T})$, where $d$ is the dimensionality of the feature space, $H$ is the horizon length, and $T$ is the total number of episodes. Our regret bound is optimal for linear (contextual) bandits which is a special case of our setting with $H = 1$.

Existing policy iteration algorithms under the same setting heavily rely on repeatedly sampling the same state by access to the simulator, which is not implementable in the online setting with stochastic initial states studied in this paper. In contrast, our new algorithm circumvents this limitation by strategically using only high-confidence part of the trajectory data and freezing the policy for well-explored states, which ensures that all data used by our algorithm remains effectively *on-policy* during the whole course of learning. We further demonstrate the versatility of our approach by extending it to the Uniform-PAC setting and to function classes with bounded eluder dimension.

## 1 Introduction

In modern reinforcement learning (RL), function approximation schemes are often employed to handle large state spaces. During the past decade, significant progress has been made towards understanding the theoretical foundation of RL with function approximation. By now, a rich set of tools and techniques have been developed (Jiang et al., 2017; Sun et al., 2019; Jin et al., 2020; Wang et al., 2020; Du et al., 2021; Jin et al., 2021; Foster et al., 2021), which have led to statistically efficient RL with function approximation under various structural assumptions. However, despite these advancements, a remaining challenge is that many of these statistically efficient RL algorithms are computationally inefficient, which severely limits the practical applicability of these approaches. Recently, these computational-statistical gaps have drawn significant interest from the RL theory community, leading to computationally efficient algorithms (Zanette et al., 2020b; Wu et al., 2024; Golowich & Moitra, 2024) and computational hardness results (Kane et al., 2022).

For the special case of RL with linear function approximation, a structural assumption common in the literature is the linear bellman completeness assumption (Zanette et al., 2020a;b; Wu et al., 2024; Golowich & Moitra, 2024), which, roughly speaking, assumes that the Bellman backups of linear state-action value functions are still linear w.r.t. a fixed feature representation given to the

learner. This assumption naturally arises when analyzing value-iteration type RL algorithms (e.g., FQI (Munos, 2005)) with linear function approximation. It is by now well-understood that RL with linear bellman completeness is statistically tractable (Zanette et al., 2020a; Du et al., 2021; Jin et al., 2021), in the sense that a near-optimal policy can be learned with polynomial sample complexity. However, the algorithm employed to achieve such polynomial sample complexity require solving computationally intractable optimization problems, and recently a number of algorithms have been proposed to fill this computational-statistical gap, leading to statistically and computationally efficient RL under linear bellman completeness for MDPs satisfying the explorability assumption (Zanette et al., 2020b), MDPs with deterministic dynamics, stochastic rewards and stochastic initial states (Wu et al., 2024), and MDPs with constant number of actions (Golowich & Moitra, 2024).

A major drawback of the linear bellman completeness assumption, as observed by Chen & Jiang (2019), is that it is *not monotone* in the function class under consideration. For the linear function case, adding more features into the feature representation may break the assumption. This makes the bellman completeness much less desirable for practical scenarios, as in modern RL applications, neural networks with billions of parameters are employed as the function approximators, and feature selection could be intractable in such cases.

For RL with linear function approximation, another common assumption is the linear $Q^\pi$ realizability (Du et al., 2019a; Lattimore et al., 2020; Yin et al., 2022; Weisz et al., 2022; 2023), which assumes that any policy's $Q$-function is linear in a fixed state-action feature representation given to the learner. This assumption naturally arises when analyzing policy-iteration type RL algorithms (e.g., LSPI (Lagoudakis & Parr, 2003)) with linear function approximation, and it has the desired monotonicity property, in the sense that adding more features never hurts realizability. However, unlike linear bellman completeness, the computational-statistical gap under linear $Q^\pi$ realizability is largely unexplored. Under this assumption, the first method with polynomial sample complexity is due to Weisz et al. (2023), and similar to linear bellman completeness, existing algorithms (Weisz et al., 2023; Mhammedi, 2025) with such sample complexity requires either computationally intractable optimization problems or oracles. Other algorithms (Du et al., 2019a; Lattimore et al., 2020; Yin et al., 2022; Weisz et al., 2022) with polynomial running time further require (local) access to a simulator, which enables the algorithm to restart from any visited states. In contrast, in the standard online RL setting, there is no known RL algorithm that is both statistically and computationally efficient. In fact, under the linear $Q^\pi$ realizability, even when the transition dynamics are deterministic, it is unclear if computationally efficient RL is possible.

In terms of methodology, the algorithm by Weisz et al. (2023) employs the *global optimism* approach and relies on maintaining large and complex version spaces. It is unclear how to implement such approach in a computationally efficient manner, and obtaining an algorithm with polynomial running time under linear $Q^\pi$ realizability was left as an open problem by Weisz et al. (2023). Mhammedi (2025) achieves sample and oracle efficient under linear $Q^\pi$ realizability by relying on a cost-sensitive classification oracle. However, such oracle could be NP-hard to implement in the worst case. Meanwhile, the algorithms by Yin et al. (2022); Weisz et al. (2022) employ the standard approximate policy-iteration framework and assume local access to a simulator. In these algorithms, before adding a state-action pair $(s, a)$ into the dataset, the algorithms use multiple rollouts starting from $(s, a)$ to ensure all successor states are well-explored, so that the policy value associated with $(s, a)$ is accurate. Once a new under-explored state appears, the whole algorithm restarts from that state and perform additional rollouts to make it well-explored. Consequently, these algorithms critically rely on the simulator to revisit the same state for multiple times. However, such resampling mechanism is not implementable in the standard online RL setting when the state-space is large. Even with deterministic dynamics, as long as the initial state is stochastic, we might not even encounter the same state twice during the whole learning process.

In this paper, we provide the first computationally efficient algorithm under the linear $Q^\pi$ realizability assumption with stochastic initial states, stochastic rewards and deterministic dynamics in the online RL setting. Our algorithm, named *Frozen Policy Iteration* (FPI), achieves a regret bound of $\widetilde{\mathcal{O}}(\sqrt{d^2 H^6 T})$[1], where $d$ is the dimensionality of the feature space, $H$ is the horizon length, and $T$ is the total number of episodes, which is optimal for linear (contextual) bandits (Dani et al., 2008), a special case with $H = 1$. Unlike existing policy iteration algorithms (Du et al., 2019a; Lattimore

---

[1]Throughout the paper, we use $\widetilde{\mathcal{O}}(\cdot)$ to suppress logarithm factors.

et al., 2020; Yin et al., 2022; Weisz et al., 2022), our new algorithm circumvents the resampling issue by strategically using only high-confidence part of the trajectory data and *freezing* the policy for well-explored states, which ensures that all data used by our algorithm remains effectively *on-policy* during the whole course of learning. We further demonstrate the versatility of our approach by extending it to the Uniform-PAC setting (Dann et al., 2017) and to function classes with bounded eluder dimension (Russo & Van Roy, 2013). Due to the simplicity of our algorithm, we are able to give a proof-of-concept implementation on standard control tasks to illustrate practicality and to ablate the role of freezing.

## 2 RELATED WORK

**Linear $Q^\pi$ Realizability.** The connection and difference between our work and prior work under Linear $Q^\pi$ Realizability (Du et al., 2019a; Lattimore et al., 2020; Yin et al., 2022; Weisz et al., 2022; 2023) have already been discussed in the introduction. In Table 1, we compare our new result with prior works to make the difference clear. We also note that the algorithms by Lattimore et al. (2020); Yin et al. (2022); Weisz et al. (2022) work in the discounted setting, while our new algorithm and the algorithm by Du et al. (2019a); Weisz et al. (2023) work under the finite-horizon setting.

**Linear $Q^*$ Realizability.** The Linear $Q^*$ Realizability assumes that the optimal $Q$-function is linear in a given state-action feature representation. Since Linear $Q^\pi$ Realizability implies Linear $Q^*$ Realizability, algorithms under Linear $Q^*$ Realizability could also be relevant to our setting. However, existing algorithms that work under this assumption either require lower bounded suboptimality gap and deterministic dynamics with deterministic initial states (Du et al., 2019b; 2020), fully deterministic system (i.e., deterministic rewards and initial states) (Wen & Van Roy, 2013), or lower bounded suboptimality gap with local access to a simulator (Li et al., 2021). All these assumptions are much stronger than those assumed by prior algorithms under Linear $Q^\pi$ Realizability.

**Uniform-PAC.** Dann et al. (2017) propose the Uniform-PAC setting as a bridge between the PAC setting and the regret minimization setting, and develop the first tabular RL algorithm with Uniform-PAC guarantees. Uniform-PAC guarantees are also achieved for RL with function approximation (He et al., 2021; Wu et al., 2023). To achieve the Uniform-PAC guarantee, we use an accuracy level framework similar to that of He et al. (2021), though the details are substantially different since our algorithm is based on policy-iteration, while their algorithm is based on value-iteration.

| | Access model | Stochastic transitions | PAC guarantee | Regret guarantee |
|---|---|---|---|---|
| Du et al. (2019a) | generative model | ✓ | ✓ | × |
| Lattimore et al. (2020) | generative model | ✓ | ✓ | × |
| Yin et al. (2022) | local access | ✓ | ✓ | × |
| Weisz et al. (2022) | local access | ✓ | ✓ | × |
| Weisz et al. (2023) | online RL (computationally intractable) | ✓ | ✓ | × |
| This work | online RL | × | ✓ | ✓ |

Table 1: Comparison with prior works.

## 3 PRELIMINARIES

A finite-horizon Markov Decision Process (MDP) is defined by the tuple $(\mathcal{S}, \mathcal{A}, H, P, R, \mu)$. $H \in \mathbb{N}^+$ is the horizon. $\mathcal{S} = \mathcal{S}_1 \cup \cdots \cup \mathcal{S}_H$ is the state space, partitioned by the horizon $H$, with $\mathcal{S}_h$ denoting the set of states at stage $h \in [H] := \{1, \cdots, H\}$. We assume that $\mathcal{S}_1, \cdots, \mathcal{S}_H$ are disjoint sets. $\mathcal{A}$ is the action space. $P : \mathcal{S}_h \times \mathcal{A} \to \Delta(\mathcal{S}_{h+1})$ is the transition function, where $\Delta(\mathcal{S}_{h+1})$ denotes the set of probability distributions over $\mathcal{S}_{h+1}$. $R : \mathcal{S} \times \mathcal{A} \to \Delta([0, 1])$ is the reward distribution, where $R(s, a)$ is a distribution on $[0, 1]$ with mean $r(s, a)$, indicating the stochastic reward received by taking action $a$ on state $s$. $\mu \in \Delta(\mathcal{S}_1)$ is the probability distribution of the initial

state. Given a policy $\pi : \mathcal{S} \to \mathcal{A}$, for each $s \in \mathcal{S}_h$ and $a \in \mathcal{A}$, we define the state-action value function of policy $\pi$ as $Q^\pi(s,a) = \mathbb{E}\left[\sum_{i=h}^H r_i \,\middle|\, s_h = s, a_h = a, \pi\right]$ and the state value function as $V^\pi(s) = Q^\pi(s, \pi(s))$. Let $\pi^*$ be the optimal policy.

For an RL algorithm, define its regret in the first $T$ episodes as the sum of the suboptimality gap of the first $T$ trajectories, $\mathrm{Reg}(T) = \sum_{t=1}^T \left(V^{\pi^*}(s_1^{(t)}) - \sum_{h=1}^H r(s_h^{(t)}, a_h^{(t)})\right)$, where $s_1^{(t)}, a_1^{(t)}, \cdots, s_H^{(t)}, a_H^{(t)}$ is the trajectory in the $t$-th episode.

Let $V \in \mathbb{R}^{d \times d}$ be symmetric positive definite. For any $x \in \mathbb{R}^d$ define the elliptical norm induced by $V$ as $\|x\|_V = \sqrt{x^\top V x}$. A zero-mean random variable $X$ is called $\sigma$-subGaussian if $\mathbb{E}\left[e^{\lambda X}\right] \leq e^{\lambda^2 \sigma^2 / 2}$ for all $\lambda \in \mathbb{R}$.

In this work, we assume the Linear $Q^\pi$ Realizability assumption, formally stated below.

**Assumption 1** (Linear $Q^\pi$ Realizability). *MDP $(\mathcal{S}, \mathcal{A}, H, P, R, \mu)$ is $\kappa$-approximate $Q^\pi$ realizable with a feature map $\phi : \mathcal{S} \times \mathcal{A} \to \mathbb{R}^d$ for some $\kappa > 0$. That is, for any policy $\pi$, there exists $\theta_1^\pi, \cdots, \theta_H^\pi \in \mathbb{R}^d$ such that for any $h \in [H], s \in \mathcal{S}_h$ and $a \in \mathcal{A}$, $|Q^\pi(s,a) - \langle \phi(s,a), \theta_h^\pi \rangle| \leq \kappa$.*

We assume that the feature map $\phi$ is given to learner. Since $Q^\pi(s,a) \in [0, H]$, we also make the following assumption, which ensures that $\phi(s,a)$ and $\theta_h^\pi$ have bounded norm for all $(s,a) \in \mathcal{S} \times \mathcal{A}$ and $h \in [H]$.

**Assumption 2** (Boundedness). *For any $h \in [H]$, $\|\phi(s,a)\|_2 \leq 1$ for all $s \in \mathcal{S}_h, a \in \mathcal{A}$, and $\|\theta_h^\pi\|_2 \leq \sqrt{d}H$ for any policy $\pi$.*

We further require deterministic state transition.

**Assumption 3** (Deterministic Transitions). *For any $s \in \mathcal{S}$ and $a \in \mathcal{A}$, $P(s,a)$ is a one-point distribution, i.e., the system transitions to a unique state after taking action $a$ on state $s$.*

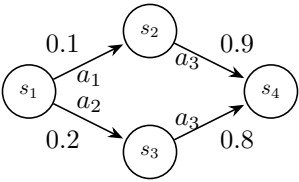

Figure 1: An example of MDP satisfying linear $Q^\pi$ realizability, where $\phi(s_1, a_1) = \phi(s_1, a_2) = e_1$, $\phi(s_2, a_3) = e_2$, $\phi(s_3, a_3) = e_3$.

We note that linear $Q^\pi$ realizability does not imply linear bellman completeness even in MDPs with deterministic transitions. Figure 1 shows an example satisfying linear $Q^\pi$ realizability which is not linear bellman complete. Moreover, although we assume the transition to be deterministic, the initial state distribution is stochastic (in fact, our algorithm allows for adversarially chosen initial states), and the reward signal could be stochastic. Therefore, learning is still challenging in this case. Also, stochastic initial state could in fact cover a number of standard RL benchmarks including classical control tasks (e.g., CartPole, Acrobot) (Towers et al., 2024) and MuJoCo Environments (Todorov et al., 2012), where randomness primarily arises from the reset distribution rather than process noise. For Atari Games, when sticky actions are disabled, the simulator yields deterministic dynamics; stochasticity is typically injected only at reset.

# 4 WARMUP: FROZEN POLICY ITERATION IN THE PAC SETTING

In this section, we present the probably approximately correct (PAC) version of our main algorithm, which serves as a warmup of the more complicated regret minimization algorithm in Section 5.

## 4.1 THE ALGORITHM

**Datasets.** In our algorithm, for each step $h \in [H]$, we maintain a dataset $\mathcal{D}_h$ which is an ordered sequence consisting of $(s_{h,i}, a_{h,i}, q_{h,i})$, where $(s_{h,i}, a_{h,i}) \in \mathcal{S}_h \times \mathcal{A}$ is a state-action pair, and

---

**Algorithm 1** Frozen Policy Iteration-PAC (FPI-PAC)

---

1: Initialize $\mathcal{D}_{1,h} = \{\}$ for all $h \in [H]$
2: **for** $t = 1, \cdots, T$ **do**
3:     For each $h \in [H]$ and $s \in \mathcal{S}_h$, define

$$\pi_t(s) = \begin{cases} \arg\max_{a \in \mathcal{A}} Q_t(s,a) & (s,a) \in \text{Cover}\,(\mathcal{D}_{t,h}, \varepsilon) \text{ for all } a \in \mathcal{A} \\ \text{any } a \in \mathcal{A} \text{ such that } (s,a) \notin \text{Cover}\,(\mathcal{D}_{t,h}, \varepsilon) & \text{otherwise} \end{cases},$$

   where $\text{Cover}\,(\mathcal{D}_{t,h}, \varepsilon)$ is as defined in (1) and $Q_t(s,a)$ is as defined in (3)
4:     Receiving $s_1^{(t)}, a_1^{(t)}, r_1^{(t)}, s_2^{(t)}, a_2^{(t)}, r_2^{(t)}, \ldots, s_H^{(t)}, a_H^{(t)}, r_H^{(t)}$ by executing $\pi_t$
5:     $h_t \leftarrow \max \left\{ h \in [H] : (s_h^{(t)}, a_h^{(t)}) \notin \text{Cover}(\mathcal{D}_{t,h}, \varepsilon) \right\}$
6:     **if** $h_t$ exists **then**
7:         $\hat{q}_t \leftarrow \sum_{h=h_t}^{H} r_h^{(t)}$
8:         Update $\mathcal{D}_{t+1,h_t} \leftarrow \mathcal{D}_{t,h_t} \cup \left\{ \left( s_{h_t}^{(t)}, a_{h_t}^{(t)}, \hat{q}_t \right) \right\}$ and $\mathcal{D}_{t+1,h} \leftarrow \mathcal{D}_{t,h}$ for any $h \neq h_t$
9:     **end if**
10: **end for**

---

$q_{h,i}$ is total reward obtained by following a policy $\pi_t$ starting from $(s_{h,i}, a_{h,i})$. We further define $\phi_{h,i} = \phi\,(s_{h,i}, a_{h,i})$ to be the feature of $(s_{h,i}, a_{h,i})$. In the description of Algorithm 1, for clarity, we use $\mathcal{D}_{t,h}$ to denote the snapshot of $\mathcal{D}_h$ *before* the $t$-th round, and we initialize $\mathcal{D}_{1,h}$, i.e., the dataset before the first round, to be the empty set for all $h \in [H]$.

**Exploration Mechanism.** For each round $t$, we first define a policy $\pi_t$, where for each state $s$ at step $h \in [H]$, we first test if for all actions $a \in \mathcal{A}$, $(s,a)$ could be covered by existing data in $\mathcal{D}_{t,h}$. Concretely, for a dataset $\mathcal{D} = \{(s_i, a_i, q_i)\}_{i=1}^n$, define

$$\text{Cover}(\mathcal{D}, \varepsilon) = \{(s,a) \in \mathcal{S} \times \mathcal{A} : \|\phi(s,a)\|_{\Sigma^{-1}} \leq \varepsilon\} \tag{1}$$

where $\Sigma = \lambda I + \sum_{i=1}^n \phi(s_i, a_i)\phi(s_i, a_i)^\top$ is the regularized empirical feature covariance matrix. Equivalently, $\text{Cover}(\mathcal{D}, \varepsilon)$ contains all state-action pairs $(s,a)$ for which the least squares estimate has error upper bounded by (roughly) $\varepsilon$. If for all actions $a \in \mathcal{A}$, $(s,a)$ lies in $\text{Cover}(\mathcal{D}, \varepsilon)$, we define $\pi_t(s)$ to be the greedy policy with respect to a $Q$-function $Q_t$ (defined later). Otherwise, there must be an action $a \in \mathcal{A}$ satisfying $(s,a) \notin \text{Cover}(\mathcal{D}, \varepsilon)$, and we define $\pi_t(s)$ to be any such $a$ to encourage exploration. We break tie in a consistent manner when there are multiple such $a$.

**Avoiding Off-Policy Data and Resampling.** After defining a policy $\pi_t$, we execute $\pi_t$ to receive a trajectory sample, forms a new dataset $\mathcal{D}_{t+1,h}$ based on $\mathcal{D}_{t,h}$ and the collected data, and proceeds to the next round $t + 1$. So far, the design of our algorithm does not differ significantly from the standard policy iteration framework. Indeed, the main novelty of Algorithm 1 lies in how we form the new dataset $\mathcal{D}_{t+1,h}$ and define a $Q$-function $Q_{t+1}$ based on $\mathcal{D}_{t+1,h}$. If not handled properly, our datasets could contain *off-policy* data once $\pi_t$ is updated. In particular, for each data $(s_{h,i}, a_{h,i}, q_{h,i})$, $q_{h,i}$ might deviate significantly from the $Q$-value of $(s_{h,i}, a_{h,i})$ once the policy is updated. We note that assuming access to a simulator (Du et al., 2019a; Lattimore et al., 2020; Yin et al., 2022; Weisz et al., 2022; 2023), one could simply recollect samples by executing the updated policy starting from $(s_{h,i}, a_{h,i})$ to estimate its $Q$-value with respect to the updated policy. Unfortunately, in the online setting with stochastic initial states, such resampling is not implementable as we might not even encounter the same state twice during the whole learning process.

**Updating Datasets.** In the design of Algorithm 1, we only include the high-confidence part of the trajectory data into the new datasets. In particular, in Step 5 of Algorithm 1, we define $h_t$ to be last step $h$, so that $(s_h^{(t)}, a_h^{(t)})$ could be not covered by existing data. We then add $(s_{h_t}^{(t)}, a_{h_t}^{(t)}, \hat{q}_t)$ into $\mathcal{D}_{t,h_t}$ to form $\mathcal{D}_{t+1,h_t}$ where $\hat{q}_t$ is the total reward starting from step $h_t$. For all other step $h \neq h_t$, we keep $\mathcal{D}_{t,h}$ unchanged. This design choice is due to the following reason: for all steps $h > h_t$, we must have $(s_h^{(t)}, a) \in \text{Cover}\,(\mathcal{D}_{t,h}, \varepsilon)$ for all $a \in \mathcal{A}$, since otherwise an exploratory action would have been chosen (cf. the definition of $\pi_t$), which further implies that $\pi_t$ is a near-optimal policy for $s_{h_t+1}^{(t)}, s_{h_t+2}^{(t)}, \ldots, s_H^{(t)}$. On the other hand, since $(s_{h_t}^{(t)}, a_{h_t}^{(t)}) \notin \text{Cover}\,(\mathcal{D}_{t,h_t}, \varepsilon)$, $a_{h_t}^{(t)}$ could be a

suboptimal action for $s_{h_t}^{(t)}$. Because of this, we only include $(s_{h_t}^{(t)}, a_{h_t}^{(t)})$ into the new dataset, and discard all other state-action pairs on the trajectory.

**Freezing High-Confidence States.** It remains to define the $Q$-function $Q_t$ based on the dataset $\mathcal{D}_{t,h}$. Here, a naïve choice is to use all data in $\mathcal{D}_{t,h}$ to obtain a least-squares estimate and define $Q_t$ accordingly, but by doing so, eventually our datasets could still contain *off-policy* data once the policy $\pi_t$ is updated. In particular, although $s_h^{(t)}$ lies in the high-confidence region coverd by $\mathcal{D}_{t,h}$ for all $h > h_t$, the policy on state $s_h^{(t)}$ could still be changed in later rounds once we update $\pi_t$, which means our dataset contains reward sums from different policies. Here, our new idea is to *freeze* the updates on $\pi_t(s)$, once our datasets could cover $(s, a)$ for all $a \in \mathcal{A}$. Concretely, for each round $t$, for each $s \in \mathcal{S}_h$, we define

$$k_t(s) = \min \left\{ k \in \mathbb{N} : (s, a) \in \text{Cover}\left(\{(s_{h,i}, a_{h,i}, q_{h,i})\}_{i=1}^k, \varepsilon\right) \text{ for all } a \in \mathcal{A} \right\} \wedge |\mathcal{D}_{t,h}|. \quad (2)$$

Recall that $\mathcal{D}_{t,h} = \{(s_{h,i}, a_{h,i}, q_{h,i})\}_{i=1}^{|\mathcal{D}_{t,h}|}$ is the snapshot of our dataset at step $h$ before the $t$-th round. Essentially, $k_t(s)$ is the first time we add a data into $\mathcal{D}_{t,h}$, after which $(s, a)$ could be covered by $\mathcal{D}_{t,h}$ for all $a \in \mathcal{A}$. Therefore, by defining $Q_t(s, \cdot)$ using only the first $k_t(s)$ data in $\mathcal{D}_{t,h}$, we effectively freeze the updates on $\pi_t(s)$ once our datasets could cover $(s, a)$ for all $a \in \mathcal{A}$. Therefore, for each $h \in [H]$ and $s \in \mathcal{S}_h$, we define

$$Q_t(s, a) = \tilde{Q}_{k_t(s)}(s, a), \quad (3)$$

where $k_t(s)$ is as defined in (2) and

$$\tilde{Q}_k(s, a) = \left\langle \phi(s, a), \Sigma_{h,k}^{-1} \sum_{i=1}^k \phi_{h,i} q_{h,i} \right\rangle$$

is the least squares estimate by using only the first $k$ data in $\mathcal{D}_h$ with

$$\Sigma_{h,k} = \lambda I + \sum_{i=1}^k \phi_{h,i} \phi_{h,i}^\top. \quad (4)$$

## 4.2 THE ANALYSIS

In this section, we outline the analysis of Algorithm 1. In our analysis, define

$$D = \frac{2d}{\varepsilon^2} \ln\left(1 + \frac{4\varepsilon^{-4}}{\lambda^2}\right). \quad (5)$$

Our first lemma shows that for any round $t$ and $h \in [H]$, $|\mathcal{D}_{t,h}|$ is always upper bounded by $D$.

**Lemma 1.** *For any $t \geq 1, h \in [H]$, it holds that $|\mathcal{D}_{t,h}| \leq D$.*

The proof of Lemma 1 is based on the observation that if we add a data $(s_{h,i}, a_{h,i}, q_{h,i})$ into $\mathcal{D}_{t+1,h}$ in the $t$-th round, we must have $(s_{h,i}, a_{h,i}) \notin \text{Cover}(\mathcal{D}_{t,h}, \varepsilon)$. Therefore, by the standard elliptical potential lemma (Lattimore & Szepesvári, 2020), Lemma 1 holds.

Our second lemma shows that for a state-action pair $(s_{h,i}, a_{h,i})$ added into $\mathcal{D}_h$ in the $t$-th round, the $Q$-function of $(s_{h,i}, a_{h,i})$ with respect to any later policy $\pi_{t'}$ (i.e., $t' \geq t$) will be unchanged.

**Lemma 2.** *For any $h \in [H]$ and $i \geq 1$, let $t$ be the round when $(s_{h,i}, a_{h,i}, q_{h,i})$ is appended to $\mathcal{D}_h$, i.e., $(s_{h,i}, a_{h,i}, q_{h,i}) \in \mathcal{D}_{t+1,h}$ while $(s_{h,i}, a_{h,i}, q_{h,i}) \notin \mathcal{D}_{t,h}$. Then $Q^{\pi_t}(s_{h,i}, a_{h,i}) = Q^{\pi_{t'}}(s_{h,i}, a_{h,i})$ for all $t' \geq t$.*

The proof of Lemma 2 is based on the following observation: suppose we add a state-action pair $(s_{h_t}^{(t)}, a_{h_t}^{(t)})$ into the dataset in the $t$-th round, for all $h > h_t$, we must have $(s_h^{(t)}, a) \in \text{Cover}(\mathcal{D}_{t,h}, \varepsilon)$ for all $a \in \mathcal{A}$, since otherwise an exploratory action $a$ with $(s_h^{(t)}, a) \notin \text{Cover}(\mathcal{D}_{t,h}, \varepsilon)$ would have been chosen. Therefore, by the way we define $Q_t$ (cf. (3)), the action of $s_h^{(t)}$ will be frozen for policies $\pi_{t'}$ defined in later rounds $t' > t$. In short, Lemma 2 formalizes the intuition that all data used by our algorithm remain effectively *on-policy* even if the policy $\pi_t$ is updated.

For each round $t \geq 1$, let $\xi_t = \hat{q}_t - Q^{\pi_t}(s_{h_t}^{(t)}, a_{h_t}^{(t)})$. Because both $\pi_t$ and the MDP transitions are deterministic, $s_h^{(t)}$ and $a_h^{(t)}$ are determined given $\pi_t$. Consequently, $\xi_t$ is the sum of at most $H$ independent 1-subGaussians, and is therefore $\sqrt{H}$-subGaussian. Our third lemma defines a high probability event, which we will condition on in later parts of the proof. Its proof is based on standard concentration inequalities for self-normalized processes (Abbasi-Yadkori et al., 2011).

**Lemma 3.** *Define event $\mathfrak{E}_{high}$ as*

$$\left\{ \forall h \in [H], k \geq 1, \left\| \sum_{i=1}^{k} \phi_{h,i} \xi_{t_{h,i}} \right\|_{\Sigma_{h,k}^{-1}}^2 \leq 2H \left( \frac{d}{2} \ln \left( 1 + \frac{k}{\lambda d} \right) + \ln \frac{H}{\delta} \right) \right\}.$$

*Then $\mathbb{P}[\mathfrak{E}_{high}] \geq 1 - \delta$.*

Define

$$\alpha = \sqrt{2H \left( \frac{d}{2} \ln \left( 1 + \frac{D}{\lambda d} \right) + \ln \frac{H}{\delta} \right)} + \sqrt{D}\kappa + \sqrt{\lambda d}H,$$

where $D$ is as defined in (5). Lemma 4 show that under the event $\mathfrak{E}_{high}$ defined in Lemma 3, $|\tilde{Q}_k(s,a) - Q^{\pi_t}(s,a)|$ is upper bounded by $\alpha \|\phi(s,a)\|_{\Sigma_{h,k}^{-1}} + \kappa$.

**Lemma 4.** *Under $\mathfrak{E}_{high}$, for any $t \geq 1, h \in [H], 0 \leq k \leq |\mathcal{D}_{t,h}|, s \in \mathcal{S}_h, a \in \mathcal{A}$, it holds that $|\tilde{Q}_k(s,a) - Q^{\pi_t}(s,a)| \leq \alpha \|\phi(s,a)\|_{\Sigma_{h,k}^{-1}} + \kappa$, where $\Sigma_{h,k}$ is as defined in (4).*

The proof of Lemma 4 uses the high probability event in Lemma 3, and also critically relies on Lemma 2 which shows that all data in our dataset remain on-policy during the whole algorithm.

The following lemma, which is a direct implication of Lemma 4 and the definition of $\mathrm{Cover}(\mathcal{D}_{t,h}, \varepsilon)$ (cf. (1)) and $Q_t$ (cf. (3)), shows that for those state-action pairs $(s,a) \in \mathrm{Cover}(\mathcal{D}_{t,h}, \varepsilon)$, we have $|Q_t(s,a) - Q^{\pi_t}(s,a)| \leq \alpha \varepsilon + \kappa$.

**Lemma 5.** *Under $\mathfrak{E}_{high}$, for any $t \geq 1, h \in [H], s \in \mathcal{S}_h, a \in \mathcal{A}$, if $(s,a) \in \mathrm{Cover}(\mathcal{D}_{t,h}, \varepsilon)$, then $|Q_t(s,a) - Q^{\pi_t}(s,a)| \leq \alpha \varepsilon + \kappa$.*

Our final lemma characterizes the suboptimality of $\pi_t$ for those well-explored states $s \in \mathcal{S}$ satisfying $(s,a) \in \mathrm{Cover}(\mathcal{D}_{t,h}, \varepsilon)$ for all $a \in \mathcal{A}$.

**Lemma 6.** *Under $\mathfrak{E}_{high}$, given $t \geq 1, s \in \mathcal{S}_1$, if $(s,a) \in \mathrm{Cover}(\mathcal{D}_{t,1}, \varepsilon)$ for all $a \in \mathcal{A}$, then we have $V^{\pi_t}(s) \geq V^{\pi^*}(s) - 2H(\alpha \varepsilon + \kappa)$.*

To prove Lemma 6, by the performance difference lemma (Kakade & Langford, 2002), we only need to show that for the state-action pair $(s_h^*, a_h^*)$ at each step $h$ of the trajectory induced by the optimal policy $\pi^*$, switching from $a_h^* = \pi^*(s_h^*)$ to $\pi_t(s_h^*)$ reduces the policy value of $\pi_t$ by at most $2(\alpha \varepsilon + \kappa)$. If for all actions $a \in \mathcal{A}$, $(s_h^*, a)$ could be covered by $\mathcal{D}_{t,h}$, then the above claim is clearly true by Lemma 5 and the definition of $\pi_t$ and $Q_t$ in (3). Otherwise, we could find a well-explored state $s_{h'}^*$ for some $h' < h$. Using the $Q^\pi$ realizability assumption (Assumption 1) for such $s_{h'}^*$, we further shows the policy value of any $a \in A$ at state $s_h^*$ differs by at most $\alpha \varepsilon + \kappa$.

By taking $\lambda = H^{-1}$ and scaling $\varepsilon$ by a factor of $\Theta(H\alpha)$ (denoted by $\bar{\varepsilon}$), we arrive at the final guarantee of Algorithm 1.

**Theorem 1.** *There is a constant $C$ so that for any $\bar{\varepsilon} > C\sqrt{d}H\kappa$, by picking certain $\varepsilon$, with probability at least $1 - \delta$, the number of episodes with suboptimality gap greater than $\bar{\varepsilon}$ is at most $\widetilde{\mathcal{O}} \left( \frac{d^2 H^4}{\bar{\varepsilon}^2} \right)$.*

To prove Theorem 1, note that if $(s_1^{(t)}, a) \in \mathrm{Cover}(D_{t,1}, \varepsilon)$ for all $a \in \mathcal{A}$, Lemma 6 guarantees that the suboptimality gap of such episode is at most $\bar{\varepsilon}$. In the other episodes, we must have increased the size of one dataset by 1, whose total occurrence count is upper bounded by $HD$ because every dataset will be updated at most $D$ times by Lemma 1.

**Time and Space Complexity.** Assume that $\phi(s,a)$ can be computed in $\mathrm{poly}(d)$ time and space given any $s \in \mathcal{S}$ and $a \in \mathcal{A}$. In Algorithm 1 the size of every dataset is bounded by $D$, so its

space complexity is $\widetilde{\mathcal{O}}\left(\frac{H \operatorname{poly}(d)}{\varepsilon^2}\right)$. We assume that the action space is finite, under which case $\pi_t(s)$ can be computed in time $\mathcal{O}(D|\mathcal{A}|\operatorname{poly}(d))$. Therefore, the time complexity of Algorithm 1 is $\widetilde{\mathcal{O}}\left(\frac{HT|\mathcal{A}|\operatorname{poly}(d)}{\varepsilon^2}\right)$.

## 5 REGRET MINIMIZATION VIA FROZEN POLICY ITERATION

In this section, we present the regret minimization version of Frozen Policy Iteration. See Algorithm 2 for the formal description.

---

**Algorithm 2** Frozen Policy Iteration-Regret Minimization (FPI-Regret)

---

1: Initialize $\mathcal{D}_h^{(l)}$ as empty list for all $l \geq 0, h \in [H]$
2: **for** $t = 1, \cdots, T$ **do**
3:      $l \leftarrow \overline{L}$                                              $\triangleright \overline{L}$ is defined in Section B.1
4:      get $s_1^{(t)}$
5:      **for** $h = 1, \cdots, H$ **do**
6:          **if** all of $\mathfrak{I}_t^{(1)}(s_h^{(t)}), \cdots, \mathfrak{I}_t^{(l)}(s_h^{(t)})$ are true **then**        $\triangleright \mathfrak{I}_t^{(l)}$ is as defined in (6)
7:              $l_h^{(t)} \leftarrow l$
8:              take action $a_h^{(t)} \leftarrow \pi_t^{(l)}(s_h^{(t)})$              $\triangleright \pi_t^{(l)}$ is as defined in (8)
9:          **else**
10:              $h_t \leftarrow h$
11:              $l \leftarrow \min\left\{l \geq 1 : \mathfrak{I}_t^{(l)}(s_h^{(t)}) \text{ is not true}\right\}$
12:              $l_h^{(t)} \leftarrow l$
13:              pick any $a \in \mathcal{A}_t^{(l)}(s_h^{(t)})$ such that $(s_h^{(t)}, a) \notin \operatorname{Cover}(\mathcal{D}_h^{(l)}, 2^{-l})$
14:              take action $a_h^{(t)} \leftarrow a$                    $\triangleright \mathcal{A}_t^{(l)}$ is as defined in (7)
15:          **end if**
16:          get $r_h^{(t)}, s_{h+1}^{(t)}$
17:      **end for**
18:      **if** $l < \overline{L}$ **then**
19:          $\hat{q}_t \leftarrow \sum_{h=h_t}^H r_h^{(t)}$
20:          $\mathcal{D}_{h_t}^{(l)}.\operatorname{append}(s_{h_t}^{(t)}, a_{h_t}^{(t)}, \hat{q}_t)$
21:      **end if**
22: **end for**

---

**Datasets and Accuracy Levels.** Algorithm 2 follows similar high-level framework as Algorithm 1. However, Algorithm 1 uses a fixed accuracy parameter $\varepsilon$, which is inadequate for achieving $\sqrt{T}$-type regret bound. Instead, Algorithm 2 uses multiple accuracy levels $1 \leq l < \overline{L}$, where level $l$ corresponds to an instance of Algorithm 1 with accuracy $\varepsilon = 2^{-l}$ and $\overline{L}$ is a fixed constant defined later (Section B.1). For each $l \geq 1$ and $h \in [H]$, we maintain a dataset $\mathcal{D}_h^{(l)}$ which is an ordered sequence consisting of $(s_{h,i}^{(l)}, a_{h,i}^{(l)}, q_{h,i}^{(l)})$. As in Section 4, for every $t \geq 1$, let $\mathcal{D}_{t,h}^{(l)}$ be the snapshot of $\mathcal{D}_h^{(l)}$ before the $t$-th round. We also write $\phi_{h,i}^{(l)} = \phi(s_{h,i}^{(l)}, a_{h,i}^{(l)})$ and $\Sigma_{h,k}^{(l)} = \lambda I + \sum_{i=1}^k \phi_{h,i}^{(l)} \phi_{h,i}^{(l)}{}^\top$.

**Adjusting Accuracy Level.** At the beginning of each episode, the accuracy level $l$ is initialized to be $\overline{L}$, i.e, the level with highest accuracy. For each step $h$, if there is an action $a$ such that $(s_h^{(t)}, a)$ cannot be covered by $\mathcal{D}_{t,h}^{(l')}$ with accuracy $2^{-l'}$ for some $l' \leq l$, then we would replace $l$ with $l'$ (Step 11), aiming at a lower accuracy of $2^{-l'}$. Concretely, for each $s \in \mathcal{S}_h$, define the indicator

$$\mathfrak{I}_t^{(l)}(s) = \mathbb{I}\left\{(s, a) \in \operatorname{Cover}(\mathcal{D}_{t,h}^{(l)}, 2^{-l}) \text{ for all } a \in \mathcal{A}_t^{(l)}(s)\right\}, \tag{6}$$

where $\mathcal{A}_t^{(l)}(s)$ is a subset of actions that will be defined later in this section. Here, $\operatorname{Cover}(\mathcal{D}_{t,h}^{(l)}, 2^{-l})$ follows the same definition as in (1). If all of $\mathfrak{I}_t^{(1)}(s_h^{(t)}), \cdots, \mathfrak{I}_t^{(l)}(s_h^{(t)})$ are true, we keep $l$ unchanged

and follows a greedy policy $\pi_t^{(l)}$ for the purpose of exploitation. Otherwise, we replace $l$ with the smallest $l'$ so that $\mathfrak{I}_t^{(l')}(s_h^{(t)})$ is false, and take an exploratory action $a$. Once an episode finished, we add a data $(s_{h_t}^{(t)}, a_{h_t}^{(t)}, \hat{q}_t)$ into $\mathcal{D}_{h_t}^{(l)}$, where $h_t$ is the last level where we take an exploratory action, and $\hat{q}_t$ is the sum of rewards starting from step $h_t$ on the trajectory, and proceed to the next round.

**Exploration under Accuracy Level Constraints.** A subtlety in Algorithm 2 is that how the exploratory action is chosen. Unlike Algorithm 1 where any action $a$ satisfying $(s, a) \notin \mathrm{Cover}(\mathcal{D}_{t,h}, \varepsilon)$ could be chosen, here we also need to make sure that the suboptimality incurred by the chosen exploratory action is upper bounded by (roughly) $2^{-l}$. To this end, for the $t$-th round, for each state $s$ and accuracy level $l$, we define a subset of actions, $\mathcal{A}_t^{(l)}(s)$, which include all actions with suboptimality upper bounded by $2^{-l}$. To estimate the suboptimality of actions, we use the estimated $Q$-value at accuracy level $l - 1$. Finally, when deciding whether to the freeze the policy of a state $s$ at the $t$-th round and accuracy level $l$, we only test whether $(s, a)$ could be covered by the dataset for those actions in $\mathcal{A}_t^{(l)}(s)$, instead of the whole action space $\mathcal{A}$.

Formally, we define $\mathcal{A}_t^{(l)}$, $k_t^{(l)}$ and $Q_t^{(l)}$ inductively on $l$ as follows:

$$\mathcal{A}_t^{(l)}(s) = \begin{cases} \mathcal{A} & l = 1 \\ \left\{ a \in \mathcal{A}_t^{(l-1)}(s) : Q_t^{(l-1)}(s, a) \geq Q_t^{(l-1)}(s, \pi_t^{(l-1)}(s)) - 2H\Delta_{l-1} \right\} & l > 1 \end{cases} ; \quad (7)$$

$$k_t^{(l)}(s) = \min \left\{ k \in \mathbb{N} : (s, a) \in \mathrm{Cover}\left( \left\{ (s_{h,i}^{(l)}, a_{h,i}^{(l)}) \right\}_{i=1}^{k}, 2^{-l} \right) \text{ for all } a \in \mathcal{A}_t^{(l)}(s) \right\} \wedge \left| \mathcal{D}_{t,h}^{(l)} \right| ;$$

$$Q_t^{(l)}(s, a) = \tilde{Q}_{k_t^{(l)}(s)}^{(l)}(s, a);$$

$$\pi_t^{(l)}(s) = \underset{a \in \mathcal{A}_t^{(l)}(s)}{\arg\max} \, Q_t^{(l)}(s, a), \quad (8)$$

where $\Delta_l$ is defined in Section B.1 and $\tilde{Q}_k^{(l)}(s, a) = \left\langle \phi(s, a), \Sigma_{h,k}^{(l)-1} \sum_{i=1}^k \phi_{h,i}^{(l)} q_{h,i}^{(l)} \right\rangle$ is the least squares estimate by using only the first $k$ data in $\mathcal{D}_h^{(l)}$ as in Section 4.

The formal guarantee of Algorithm 2 is summarized as the following theorem.

**Theorem 2.** *With probability $1 - \delta$, for any $T \geq 1$, the regret incurred by Algorithm 2 satisfies* $\mathrm{Reg}(T) = \widetilde{\mathcal{O}}\left( \sqrt{d^2 H^6 T} + \sqrt{d} H^2 T \kappa \right)$.

Theorem 2 is proved using a series of lemmas similar to those in Section 4, though the proofs here are more complicated due to the use of multiple accuracy levels. See Section B for the details.

**Time and Space Complexity.** In Algorithm 2, at most one dataset will be updated at each round, so its space complexity is $\mathcal{O}(T \, \mathrm{poly}(d))$. Note that $l_h^{(t)}$ is upper bounded by $t$, so we can replace Step 3 with $l \leftarrow \min\{t, \overline{L}\}$, which ensures that $\overline{L}$ does not affect the actual computational time. The time complexity of Algorithm 2 is $\widetilde{\mathcal{O}}(HT^2 |\mathcal{A}| \, \mathrm{poly}(d))$.

**Extensions.** Algorithm 2 also enjoys a Uniform-PAC guarantee of $\widetilde{\mathcal{O}}\left( \frac{d^2 H^6}{\varepsilon^2} \right)$. See Section B.6 for the details. Moreover, we further extend our results from linear function approximation to function classes with bounded eluder dimension. We provide bounds on $\mathrm{Reg}(T)$ of

$$\mathcal{O}\left( \sqrt{H^6 T \dim_E\left(\mathcal{F}, T^{-1}\right) \left( \log \frac{HT}{\delta} + \log N(\mathcal{F}, T^{-2}, \|\cdot\|_\infty) \right)} \right).$$

See Section C for the details.

## 6 EXPERIMENTS

We implement Algorithm 1 and conduct experiments on simple RL environments from the OpenAI gym. Our experiments are performed on CartPole-v1 and InvertedPendulum-v4. We use the tile

coding method (Sutton & Barto, 2018) to produce a feature map in a continuous space. The number of tiles per dimension is set to 4. For InvertedPendulum-v4, we discretize the action space into 4 actions and set the number of steps per episode to 60. In the implementation, $\lambda$ is set to $10^{-3}$ and $\varepsilon$ is set to 1. We maintain the covariance matrix $\Sigma_{h,k}$ to compute policy $\pi_t$. Since storing $\Sigma_{h,k}$ for all $h \in [H]$ and $k \leq |\mathcal{D}_{t,h}|$ would lead to a huge memory cost, we only keep the last 20 $\Sigma_{h,k}$'s for each $h \in [H]$.

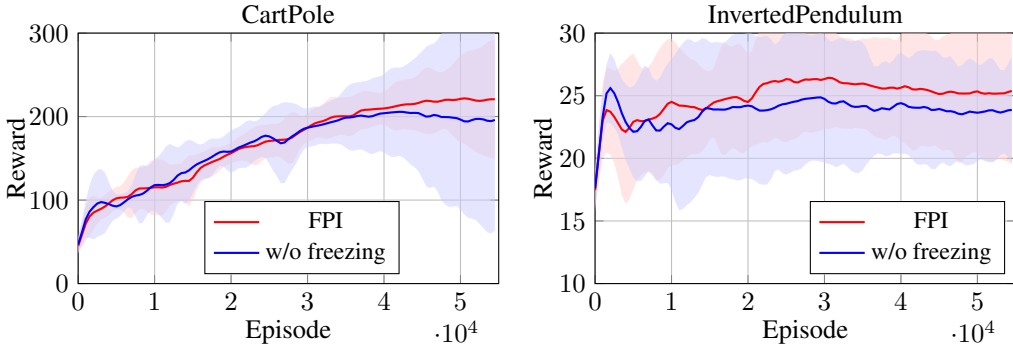

Figure 2: Learning curves on CartPole-v1 and InvertedPendulum-v4. Results are averaged over 5 seeds. The shaded area captures a 95% confidence interval around the average performance.

To ablate the role of freezing, we also implement a version of Algorithm 1 without the freezing operation, i.e., using the whole dataset to estimate $Q$-values for each step $h \in [H]$. Figure 2 shows that the freezing operation has indeed improved the performance of our algorithm.

## 7 DISCUSSIONS AND OPEN PROBLEMS

**$H$ Factors in the Regret and Uniform-PAC Bounds.** The relatively high polynomial dependence on $H$ in the regret and Uniform-PAC bounds primarily arises from the need for exploration under multiple accuracy level constraints. Ensuring that the optimal action of each state is preserved across all accuracy levels introduces additional $H$ factors compared to the PAC analysis. Further improvements on the dependence of $H$ will be an interesting future work.

**Reliance on Deterministic Transitions.** Applying our algorithm to MDPs with stochastic transitions is non-trivial, as the current analysis critically relies on the property that the state-action pairs collected in the datasets are well-explored. This property ensures that once a pair $(s, a)$ is added to the dataset, the subsequent trajectory remains within the high-confidence region, preserving the effectively on-policy nature of the data. Under stochastic dynamics, observing only one trajectory from a given pair $(s, a)$ does not guarantee that a sufficiently large or high-probability subset of trajectories starting from $(s, a)$ lies within the high-confidence region. Extending our algorithm to stochastic transition models hence remains an open problem.

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

## THE USE OF LARGE LANGUAGE MODELS

In the preparation of this paper, we employed large language models (LLMs) as a general-purpose assistive tool for non-substantive tasks. Specifically, LLMs were used for:

- Assisting in editing, proofreading, and formatting of textual content.
- Generating code snippets for creating figures and visualizations, which were then verified and customized by the authors.

The LLMs did not contribute to the core ideation, analysis, or substantive writing of the research. The authors take full responsibility for the entire content of this paper, including any portions influenced by LLMs, and affirm that the use of LLMs complies with academic integrity standards. LLMs are not considered authors or contributors to this work.

## A   MISSING PROOFS IN SECTION 4

The proofs of the lemmas in Section 4 is almost the same with the ones in Section 5.

See lemmas 8, 10, 13, 15, 16 and 20 for the proofs of lemmas 1 to 6, respectively.

## B   MISSING PROOFS IN SECTION 5

### B.1   NOTATIONS

Given any $\delta \in (0,1)$, define constants

$$\lambda = H^{-1}$$

$$\overline{L} = \left\lfloor \ln \frac{\sqrt{H}}{\kappa} \right\rfloor + 1 \ (\text{or } +\infty \text{ if } \kappa = 0)$$

$$D_l = 2d \cdot 2^{2l} \ln \left(1 + \frac{2^{4l+2}}{\lambda^2}\right) \qquad\qquad = \Theta\left(dl \cdot 2^{2l}\right)$$

$$\alpha_l = \sqrt{2H\left(\frac{d}{2}\ln\left(1+\frac{D_l}{\lambda d}\right)+\ln\frac{2Hl^2}{\delta}\right)} + \sqrt{D_l}\kappa + \sqrt{\lambda}dH \quad = \mathcal{O}\left(\sqrt{dHl\log\frac{H}{\delta}} + 2^l\sqrt{dl}\kappa\right)$$

$$\Delta_l = \alpha_l \cdot 2^{-l} + \kappa \qquad\qquad = \mathcal{O}\left(2^{-l}\sqrt{dHl\log\frac{H}{\delta}} + \sqrt{dl}\kappa\right)$$

For every $l \geq 1, h \in [H], i \geq 1$, let $t_{h,i}^{(l)}$ be the round when $(s_{h,i}^{(l)}, a_{h,i}^{(l)}, q_{h,i}^{(l)})$ is appended to $\mathcal{D}_h^{(l)}$. For every round $t \geq 1$, let $D_{t,h}^{(l)}$ be the size of $\mathcal{D}_{t,h}^{(l)}$ and $\xi_t$ be $\hat{q}_t - Q^{\pi_t^{(l)}}(s_{h_t}^{(t)}, a_{h_t}^{(l)})$.

### B.2   UPPER BOUND FOR THE SIZE OF DATASETS

**Lemma 7.** *For any $l \geq 1, h \in [H], i \geq 1$, it holds that $\left\|\phi_{h,i}^{(l)}\right\|_{\Sigma_{h,i-1}^{(l)}{}^{-1}} \geq 2^{-l}$.*

*Proof.* Denote $t_{h,i}^{(l)}$ by $t$. From algorithm 2

$$\left(s_h^{(t)}, a_h^{(t)}\right) \notin \text{Cover}\left(\mathcal{D}_{t,h}^{(l)}, 2^{-l}\right)$$

so we have

$$\left\|\phi_{h,i}^{(l)}\right\|_{\Sigma_{h,i-1}^{(l)}{}^{-1}} = \left\|\phi\left(s_h^{(t)}, a_h^{(t)}\right)\right\|_{\Sigma_{h,D_{t,h}^{(l)}}{}^{-1}} \geq 2^{-l}$$

$\square$

**Lemma 8.** *For any $t \geq 1, l \geq 1, h \in [H]$, it holds that $D_{t,h}^{(l)} \leq D_l$.*

*Proof.* Denote $D_{t,h}^{(l)}$ by $D$. By lemma 7, $2^l \left\|\phi_{h,i}^{(l)}\right\|_{\Sigma_{h,i-1}^{(l)}{}^{-1}} \geq 1$ for all $1 \leq i \leq D$, so we have

$$D \leq \sum_{i=1}^{D} \min\left(1, 2^{2l}\left\|\phi_{h,i}^{(l)}\right\|^2_{\Sigma_{h,i-1}^{(l)}{}^{-1}}\right)$$

$$\leq 2^{2l} \sum_{i=1}^{D} \min\left(1, \left\|\phi_{h,i}^{(l)}\right\|_{\Sigma_{h,i-1}^{(l)}}^{2}{}^{-1}\right)$$

$$\leq 2^{2l} \cdot 2\ln\left(\frac{\det \Sigma_{h,D}^{(l)}}{\det \Sigma_{h,0}^{(l)}}\right) \qquad \text{(elliptical potential lemma (Lattimore \& Szepesvári, 2020))}$$

$$\leq 2d \cdot 2^{2l} \ln\left(1 + \frac{D}{\lambda d}\right) \tag{9}$$

$$\leq 2d \cdot 2^{2l}\sqrt{\frac{D}{\lambda d}} \qquad (\ln(1+x) \leq \sqrt{x} \text{ for } x \geq 0)$$

Therefore, $D \leq \frac{2^{4l+2}\cdot d}{\lambda}$. Plugging it into (9) gives the desired result. $\qquad\square$

### B.3 Avoiding resampling

**Lemma 9.** *Given $t \geq 1, l \geq 1, h \in [H], s \in \mathcal{S}_h$, if all of $\mathfrak{I}_t^{(1)}(s), \cdots, \mathfrak{I}_t^{(l)}(s)$ are true, then for any $t' \geq t$,*

$$Q_{t'}^{(l)}(s,a) = Q_t^{(l)}(s,a), \forall a \in \mathcal{A}_{t'}^{(l)}(s)$$
$$\pi_{t'}^{(l)}(s) = \pi_t^{(l)}(s) \tag{10}$$
$$\mathcal{A}_{t'}^{(l+1)}(s) = \mathcal{A}_t^{(l+1)}(s)$$

*Proof.* Assume by induction (10) holds for $l = l_0 - 1$. Since $\mathfrak{I}_t^{(l)}(s)$ is true, $k_{t'}^{(l)}(s) = k_t^{(l)}(s) \leq D_{t,h}^{(l)}$ for all $t' \geq t, a \in \mathcal{A}_t^{(l)}(s)$. Note that $Q_{t'}^{(l)}(s,a)$ depends only on $k_{t'}^{(l)}(s)$; $\pi_{t'}^{(l)}(s)$ depends only on $\mathcal{A}_{t'}^{(l)}(s)$ and $Q_{t'}^{(l)}(s,a)$; and $\mathcal{A}_{t'}^{(l+1)}(s)$ depends only on $\mathcal{A}_{t'}^{(l)}(s)$ and $Q_{t'}^{(l)}(s,a)$; so (10) holds also for $l = l_0$. $\qquad\square$

**Lemma 10.** *For any $l \geq 1, h \in [H], i \geq 1, Q^{\pi_t^{(l)}}(s_{h,i}^{(l)}, a_{h,i}^{(l)}) = Q^{\pi_{t'}^{(l)}}(s_{h,i}^{(l)}, a_{h,i}^{(l)})$ for all $t' \geq t$, where $t = t_{h,i}^{(l)}$.*

*Proof.* From algorithm 2 we have $s_h^{(t)} = s_{h,i}^{(l)}, a_h^{(t)} = a_{h,i}^{(l)}$, and $\mathfrak{I}_t^{(l')}(s_{h'}^{(t)})$ is true for all $1 \leq l' \leq l, h < h' \leq H$. By lemma 9, $\pi_{t'}^{(l)}(s_{h'}^{(t)}) = \pi_t^{(l)}(s_{h'}^{(t)})$ for all $h < h' \leq H$, so the trajectory starting from $(s_h^{(t)}, a_h^{(t)})$ using policy $\pi_t^{(l)}$ is the same with the one using policy $\pi_{t'}^{(l)}$. $\qquad\square$

### B.4 Concentration bound for Q-values

Let $\{\mathcal{F}_t\}_{t=0}^{\infty}$ be a filtration such that $\mathcal{F}_t$ is the $\sigma$-algebra generated by $(r_h^{(i)})_{1 \leq i \leq t, h \in [H]}$. Then $\pi_t^{(l)}(s)$ is $\mathcal{F}_{t-1}$-measurable for all $t \geq 1, l \geq 1, s \in \mathcal{S}$. Denote $\mathbb{E}[\cdot|\mathcal{F}_t]$ by $\mathbb{E}_t[\cdot]$.

**Lemma 11.** *$\{\xi_t\}_{t=1}^{\infty}$ is a martingale difference sequence w.r.t. $\{\mathcal{F}_t\}_{t=0}^{\infty}$. Moreover, $\xi_t|\mathcal{F}_{t-1}$ is $\sqrt{H}$-subGaussian for all $t \geq 1$.*

*Proof.* Fix any $t \geq 1$. Denote $l_{h_t}^{(t)}$ by $l$. From algorithm 2 we have $a_h^{(t)} = \pi_t^{(l)}(s_h^{(t)})$ for all $h_t < h \leq H$, so

$$\mathbb{E}_{t-1}\left[\sum_{h=h_t}^{H} r_h^{(t)} \middle| h_t\right] = Q^{\pi_t^{(l)}}\left(s_{h_t}^{(t)}, a_{h_t}^{(t)}\right)$$

By tower rule, we have $\mathbb{E}_{t-1}[\xi_t] = \mathbb{E}_{t-1}\left[\sum_{h=h_t}^{H} r_h^{(t)} - Q^{\pi_t^{(l)}}(s_{h_t}^{(t)}, a_{h_t}^{(t)}) \middle| h_t\right] = 0$. Moreover, since $s_h^{(t)}$ and $a_h^{(t)}$ are $\mathcal{F}_{t-1}$ measurable for all $h \in [H]$ by the determinism of the MDP transitions, $\xi_t|\mathcal{F}_{t-1}$ is the sum of at most $H$ independent 1-subGaussians, and is therefore $\sqrt{H}$-subGaussian. $\qquad\square$

**Lemma 12** (Concentration of Self-Normalized Processes (Abbasi-Yadkori et al., 2011)). *Let $\{\varepsilon_t\}_{t=1}^{\infty}$ be a martingale difference sequence w.r.t. $\{\mathcal{F}_t\}_{t=0}^{\infty}$. Let $\varepsilon_t | \mathcal{F}_{t-1}$ be $\sigma$-subGaussian. Let $\{\phi_t\}_{t=1}^{\infty}$ be an $\mathbb{R}^d$-valued stochastic process such that $\phi_t$ is $\mathcal{F}_{t-1}$-measurable. Let $\Lambda_0$ be a $d \times d$ positive definite matrix, and $\Lambda_t$ be $\Lambda_0 + \sum_{i=1}^{t} \phi_i \phi_i^{\top}$. Then for any $\delta > 0$, with probability at least $1 - \delta$, we have for all $t \geq 1$,*

$$\left\| \sum_{i=1}^{t} \phi_i \varepsilon_i \right\|_{\Lambda_t^{-1}}^2 \leq 2\sigma^2 \ln \left( \frac{\det(\Lambda_t)^{1/2} \det(\Lambda_0)^{-1/2}}{\delta} \right)$$

**Lemma 13.** *Define event $\mathfrak{E}_{high}$ as*

$$\left\{ \forall l \geq 1, h \in [H], k \geq 1, \left\| \sum_{i=1}^{k} \phi_{h,i}^{(l)} \xi_{t_{h,i}^{(l)}}^{(l)} \right\|_{\Sigma_{h,k}^{(l)}{}^{-1}}^2 \leq 2H \left( \frac{d}{2} \ln \left( 1 + \frac{k}{\lambda d} \right) + \ln \frac{2Hl^2}{\delta} \right) \right\}$$

*Then $\mathbb{P}[\mathfrak{E}_{high}] \geq 1 - \delta$.*

*Proof.* For any $l \geq 1, h \in [H]$, $\{\xi_{t_{h,i}^{(l)}}^{(l)}\}_{i=1}^{\infty}$ is a subsequence of $\{\xi_t\}_{t=1}^{\infty}$. By lemmas 11 and 12, with probability at least $1 - \frac{\delta}{2Hl^2}$, we have for all $k \geq 1$,

$$\left\| \sum_{i=1}^{k} \phi_{h,i}^{(l)} \xi_{t_{h,i}^{(l)}}^{(l)} \right\|_{\Sigma_{h,k}^{(l)}{}^{-1}}^2 \leq 2H \ln \left( \frac{\det \left( \Sigma_{h,k}^{(l)} \right)^{1/2} \det \left( \Sigma_{h,0}^{(l)} \right)^{-1/2}}{\delta / (2Hl^2)} \right)$$

$$\leq 2H \left( \frac{d}{2} \ln \frac{\text{trace} \left( \Sigma_{h,k}^{(l)} \right)}{\lambda d} + \ln \frac{2Hl^2}{\delta} \right)$$

$$\leq 2H \left( \frac{d}{2} \ln \left( 1 + \frac{k}{\lambda d} \right) + \ln \frac{2Hl^2}{\delta} \right)$$

A union bound over $l$ and $h$ concludes the proof. $\square$

**Lemma 14** (Zanette et al. (2020a), Lemma 8). *Let $\{\delta_i\}_{i=1}^{n}$ be a real-valued sequence such that $|\delta_i| \leq \kappa$, and $\{\phi_i\}_{i=1}^{n}$ be an $\mathbb{R}^d$-valued sequence. For $\Lambda = \lambda I + \sum_{i=1}^{n} \phi_i \phi_i^{\top}$ we have*

$$\left\| \sum_{i=1}^{n} \phi_i \delta_i \right\|_{\Lambda^{-1}}^2 \leq n\kappa^2$$

**Lemma 15.** *Under $\mathfrak{E}_{high}$, for any $t \geq 1, l \geq 1, h \in [H], 0 \leq k \leq D_{t,h}^{(l)}, s \in \mathcal{S}_h, a \in \mathcal{A}$, it holds that*

$$\left| \tilde{Q}_k^{(l)}(s,a) - Q^{\pi_t^{(l)}}(s,a) \right| \leq \alpha_l \|\phi(s,a)\|_{\Sigma_{h,k}^{(l)}{}^{-1}} + \kappa$$

*Proof.* Let $\delta_i$ be $Q^{\pi_t^{(l)}} \left( s_{h,i}^{(l)}, a_{h,i}^{(l)} \right) - \left\langle \phi_{h,i}^{(l)}, \theta_h^{\pi_t^{(l)}} \right\rangle$. Then by assumption $|\delta_i| \leq \kappa$.

$$\left| \tilde{Q}_k^{(l)}(s,a) - Q^{\pi_t^{(l)}}(s,a) \right|$$

$$\leq \left| \tilde{Q}_k^{(l)}(s,a) - \left\langle \phi(s,a), \theta_h^{\pi_t^{(l)}} \right\rangle \right| + \left| \left\langle \phi(s,a), \theta_h^{\pi_t^{(l)}} \right\rangle - Q^{\pi_t^{(l)}}(s,a) \right|$$

$$\leq \left| \tilde{Q}_k^{(l)}(s,a) - \left\langle \phi(s,a), \theta_h^{\pi_t^{(l)}} \right\rangle \right| + \kappa$$

$$\tilde{Q}_k^{(l)}(s,a) - \left\langle \phi(s,a), \theta_h^{\pi_t^{(l)}} \right\rangle$$

$$= \left\langle \phi(s,a), \Sigma_{h,k}^{(l)}{}^{-1} \sum_{i=1}^{k} \phi_{h,i}^{(l)} q_{h,i}^{(l)} - \theta_h^{\pi_t^{(l)}} \right\rangle$$

$$
= \left\langle \phi(s,a), \Sigma_{h,k}^{(l)}{}^{-1} \sum_{i=1}^{k} \phi_{h,i}^{(l)} \left( Q^{\pi_{t_{h,i}^{(l)}}^{(l)}} \left( s_{h,i}^{(l)}, a_{h,i}^{(l)} \right) + \xi_{t_{h,i}^{(l)}} \right) - \theta_h^{\pi_t^{(l)}} \right\rangle
$$

$$
= \left\langle \phi(s,a), \Sigma_{h,k}^{(l)}{}^{-1} \sum_{i=1}^{k} \phi_{h,i}^{(l)} \left( Q^{\pi_t^{(l)}} \left( s_{h,i}^{(l)}, a_{h,i}^{(l)} \right) + \xi_{t_{h,i}^{(l)}} \right) - \theta_h^{\pi_t^{(l)}} \right\rangle \qquad \text{(lemma 10)}
$$

$$
= \left\langle \phi(s,a), \Sigma_{h,k}^{(l)}{}^{-1} \sum_{i=1}^{k} \phi_{h,i}^{(l)} \left( \phi_{h,i}^{(l)}{}^{\top} \theta_h^{\pi_t^{(l)}} + \delta_i + \xi_{t_{h,i}^{(l)}} \right) - \theta_h^{\pi_t^{(l)}} \right\rangle
$$

$$
= \left\langle \phi(s,a), \Sigma_{h,k}^{(l)}{}^{-1} \sum_{i=1}^{k} \phi_{h,i}^{(l)} \left( \delta_i + \xi_{t_{h,i}^{(l)}} \right) - \lambda \Sigma_{h,k}^{(l)}{}^{-1} \theta_h^{\pi_t^{(l)}} \right\rangle
$$

$$
\leq \|\phi(s,a)\|_{\Sigma_{h,k}^{(l)}{}^{-1}} \left( \left\| \sum_{i=1}^{k} \phi_{h,i}^{(l)} \xi_{t_{h,i}^{(l)}} \right\|_{\Sigma_{h,k}^{(l)}{}^{-1}} + \left\| \sum_{i=1}^{k} \phi_{h,i}^{(l)} \delta_i \right\|_{\Sigma_{h,k}^{(l)}{}^{-1}} + \left\| \lambda \theta_h^{\pi_t^{(l)}} \right\|_{\Sigma_{h,k}^{(l)}{}^{-1}} \right)
$$
$$
\text{(Cauchy-Schwartz)}
$$

$$
\leq \|\phi(s,a)\|_{\Sigma_{h,k}^{(l)}{}^{-1}} \left( \sqrt{2H \left( \frac{d}{2} \ln \left( 1 + \frac{k}{\lambda d} \right) + \ln \frac{2Hl^2}{\delta} \right)} + \sqrt{k} \kappa + \sqrt{\lambda} \left\| \theta_h^{\pi_t^{(l)}} \right\|_2 \right)
$$
$$
\text{(lemmas 12 and 14)}
$$

$$
\leq \alpha_l \|\phi(s,a)\|_{\Sigma_{h,k}^{(l)}{}^{-1}} \qquad \text{(assumption 2 and lemma 8)}
$$

$\square$

**Lemma 16.** *Under* $\mathfrak{E}_{high}$, *given* $t \geq 1, l \geq 1, h \in [H], s \in \mathcal{S}_h$, *if* $\mathfrak{I}_t^{(l)}(s)$ *is true, then for any* $a \in \mathcal{A}_t^{(l)}(s)$,
$$
\left| Q_t^{(l)}(s,a) - Q^{\pi_t^{(l)}}(s,a) \right| \leq \Delta_l
$$

*Proof.* Denote $k_t^{(l)}(s)$ by $k$. By definition $k \leq D_{t,h}^{(l)}$. Since $\mathfrak{I}_t^{(l)}(s)$ is true, $\|\phi(s,a)\|_{\Sigma_{h,k}^{(l)}{}^{-1}} \leq 2^{-l}$ for all $a \in \mathcal{A}_t^{(l)}(s)$, so we have

$$
\left| Q_t^{(l)}(s,a) - Q^{\pi_t^{(l)}}(s,a) \right|
$$
$$
= \left| \tilde{Q}_k^{(l)}(s,a) - Q^{\pi_t^{(l)}(s,a)} \right| \qquad \text{(by definition)}
$$
$$
\leq \alpha_l \|\phi(s,a)\|_{\Sigma_{h,k}^{(l)}{}^{-1}} + \kappa \qquad \text{(lemma 15)}
$$
$$
\leq \Delta_l
$$

$\square$

## B.5 REGRET ANALYSIS

**Lemma 17.** *Given* $\varepsilon > 0, v_0, v_1, \cdots, v_n \in \mathbb{R}^d$, *let* $\Lambda$ *be* $\lambda I + \sum_{i=1}^{n} v_i v_i^{\top}$. *If* $\|v_0\|_{\Lambda^{-1}} \leq \varepsilon$, *then there exist coefficients* $c_1, \cdots, c_n \in \mathbb{R}$ *such that* $\sum_{i=1}^{n} c_i^2 \leq \varepsilon^2$ *and* $\left\| v_0 - \sum_{i=1}^{n} c_i v_i \right\|_2 \leq \sqrt{\lambda} \varepsilon$.

*Proof.* Let $u$ be $\Lambda^{-1} v_0$. Then $v_0 = \Lambda u = \lambda u + \sum_{i=1}^{n} v_i v_i^{\top} u$. Pick $c_i = v_i^{\top} u$. We can verify that

$$
\sum_{i=1}^{n} c_i^2 = \sum_{i=1}^{n} \left( v_i^{\top} u \right)^2 \leq \lambda u^{\top} u + \sum_{i=1}^{n} \left( v_i^{\top} u \right)^2 = u^{\top} \Lambda u = \|v_0\|_{\Lambda^{-1}}^2 \leq \varepsilon^2
$$
$$
\left\| v_0 - \sum_{i=1}^{n} c_i v_i \right\|_2 = \|\lambda u\|_2 \leq \sqrt{\lambda} \|u\|_{\Lambda} \leq \sqrt{\lambda} \varepsilon
$$

$\square$

**Lemma 18.** *For $\varepsilon > 0, 1 \leq h_1 < h_2 \leq H$, given a policy $\pi_0$ and $n+1$ trajectories*

$$\left\{ \left( s_{h_1}^{(i)}, a_{h_1}^{(i)}, \cdots, s_{h_2}^{(i)}, a_{h_2}^{(i)} \right) \right\}_{i=0}^{n}$$

*such that $\pi_0(s_h^{(i)}) = a_h^{(i)}$ for all $0 \leq i \leq n, h_1 + 1 \leq h \leq h_2$, if*

$$\left( s_{h_1}^{(0)}, a_{h_1}^{(0)} \right) \in \mathrm{Cover}\left( \left\{ \left( s_{h_1}^{(i)}, a_{h_1}^{(i)} \right) \right\}_{i=1}^{n}, \varepsilon \right)$$

*and $s_{h_2}^{(0)} \neq s_{h_2}^{(i)}$ for all $1 \leq i \leq n$, then for any policy $\pi$ and any $a_1, a_2 \in \mathcal{A}$, it holds that*

$$\left| Q^\pi \left( s_{h_2}^{(0)}, a_1 \right) - Q^\pi \left( s_{h_2}^{(0)}, a_2 \right) \right| \leq 2 \left( \kappa + \sqrt{\lambda d} H \varepsilon + \sqrt{n} \kappa \varepsilon \right)$$

*Proof.* Fix a policy $\pi$. Define policy $\overline{\pi}$ and $\overline{\pi}_a$ for $a \in \mathcal{A}$ as follows, where $\mathrm{stage}(s) = h$ if $s \in \mathcal{S}_h$,

$$\overline{\pi}(s) = \begin{cases} \pi_0(s) & \mathrm{stage}(s) \leq h_2 \\ \pi(s) & \mathrm{stage}(s) > h_2 \end{cases}$$

$$\overline{\pi}_a(s) = \begin{cases} a & s = s_{h_2}^{(0)} \\ \overline{\pi}(s) & s \neq s_{h_2}^{(0)} \end{cases}$$

Since $s_{h_2}^{(0)} \neq s_{h_2}^{(i)}$ for all $1 \leq i \leq n$, we have $Q^{\overline{\pi}}(s_{h_1}^{(i)}, a_{h_1}^{(i)}) = Q^{\overline{\pi}_a}(s_{h_1}^{(i)}, a_{h_1}^{(i)})$. Let $R$ be

$$\mathbb{E}\left[ r_{h_1} + \cdots + r_{h_2 - 1} \,\Big|\, s_{h_1} = s_{h_1}^{(0)}, a_{h_1} = a_{h_1}^{(0)}, \pi_0 \right]$$

Then $Q^{\overline{\pi}_a}(s_{h_1}^{(0)}, a_{h_1}^{(0)}) = R + Q^\pi(s_{h_2}^{(0)}, a)$ for all $a \in \mathcal{A}$.

Denote $\phi(s_{h_1}^{(i)}, a_{h_1}^{(i)})$ by $\phi_i$. By lemma 17 there exist coefficients $c_1, \cdots, c_n \in \mathbb{R}$ such that $\sum_{i=1}^{n} c_i^2 \leq \varepsilon^2$ and

$$\left\| \phi_0 - \sum_{i=1}^{n} c_i \phi_i \right\|_2 \leq \sqrt{\lambda} \varepsilon$$

Let $q_0$ be $\sum_{i=1}^{n} c_i Q^{\overline{\pi}}(s_{h_1}^{(i)}, a_{h_1}^{(i)}) - R$. Then we have for any $a \in \mathcal{A}$,

$$\left| Q^\pi \left( s_{h_2}^{(0)}, a \right) - q_0 \right|$$

$$= \left| Q^\pi \left( s_{h_2}^{(0)}, a \right) + R - \sum_{i=1}^{n} c_i Q^{\overline{\pi}} \left( s_{h_1}^{(i)}, a_{h_1}^{(i)} \right) \right|$$

$$= \left| Q^{\overline{\pi}_a} \left( s_{h_1}^{(0)}, a_{h_1}^{(0)} \right) - \sum_{i=1}^{n} c_i Q^{\overline{\pi}_a} \left( s_{h_1}^{(i)}, a_{h_1}^{(i)} \right) \right|$$

$$\leq \left| Q^{\overline{\pi}_a} \left( s_{h_1}^{(0)}, a_{h_1}^{(0)} \right) - \langle \phi_0, \theta_{h_1}^{\overline{\pi}_a} \rangle \right| + \left| \left\langle \phi_0 - \sum_{i=1}^{n} c_i \phi_i, \theta_{h_1}^{\overline{\pi}_a} \right\rangle \right| + \sum_{i=1}^{n} c_i \left| \langle \phi_i, \theta_{h_1}^{\overline{\pi}_a} \rangle - Q^{\overline{\pi}_a} \left( s_{h_1}^{(i)}, a_{h_1}^{(i)} \right) \right|$$

$$\leq \kappa + \sqrt{\lambda} \varepsilon \left\| \theta_{h_1}^{\overline{\pi}_a} \right\|_2 + \sum_{i=1}^{n} c_i \kappa$$

$$\leq \kappa + \sqrt{\lambda d} H \varepsilon + \sqrt{n} \kappa \varepsilon$$

which gives the desired result. $\square$

**Lemma 19.** *For any $l \geq 1, 1 \leq h < h' \leq H, i \geq 1$, all of $\mathfrak{I}_t^{(1)}(s_{h'}^{(t)}), \cdots, \mathfrak{I}_t^{(l)}(s_{h'}^{(t)})$ are true, where $t = t_{h,i}^{(l)}$.*

*Proof.* From the algorithm we have $h = h_t$, so policy $\pi_t^{(l)}$ is used at stage $h + 1, \cdots, H$, which means all of $\mathfrak{I}_t^{(1)}(s_{h'}^{(t)}), \cdots, \mathfrak{I}_t^{(l)}(s_{h'}^{(t)})$ are true. $\square$

**Lemma 20.** *Under $\mathfrak{E}_{high}$, given $t \geq 1, l \geq 1, h \in [H], s \in \mathcal{S}_h$, if all of $\mathfrak{I}_t^{(1)}(s), \cdots, \mathfrak{I}_t^{(l)}(s)$ are true, then for any $a \in \mathcal{A}_t^{(l)}(s)$,*

$$Q^{\pi_t^{(l)}}(s, a) \geq Q^{\pi^*}(s, a) - 2(H - h)\Delta_l$$

*Furthermore, $\pi^*(s) \in \mathcal{A}_t^{(l+1)}(s)$.*

*Proof.* Assume by induction the result holds for $l = l_0 - 1$. Let the trajectory starting from $(s, a)$ using policy $\pi^*$ be $(s_h^*, a_h^*, s_{h+1}^*, a_{h+1}^*, \cdots, s_H^*, a_H^*)$, where $s_h^* = s, a_h^* = a$. For each $h_2 = h + 1, \cdots, H$,

- if $\mathfrak{I}_t^{(l)}(s_{h_2}^*)$ is true, then

$$Q^{\pi_t^{(l)}}\left(s_{h_2}^*, \pi_t^{(l)}\left(s_{h_2}^*\right)\right)$$
$$\geq Q_t^{(l)}\left(s_{h_2}^*, \pi_t^{(l)}\left(s_{h_2}^*\right)\right) - \Delta_l \qquad\qquad \text{(lemma 16)}$$
$$\geq Q_t^{(l)}\left(s_{h_2}^*, \pi^*\left(s_{h_2}^*\right)\right) - \Delta_l \qquad\qquad \text{(by the definition of } \pi_t^{(l)}\text{)}$$
$$\geq Q^{\pi_t^{(l)}}\left(s_{h_2}^*, \pi^*\left(s_{h_2}^*\right)\right) - 2\Delta_l \qquad\qquad \text{(lemma 16)}$$

- otherwise, let $h_1$ be $\max\left\{h' < h_2 : \text{all of } \mathfrak{I}_t^{(1)}(s_{h'}^*), \cdots, \mathfrak{I}_t^{(l)}(s_{h'}^*) \text{ are true}\right\}$, which always exists since $h$ satisfies the condition. Define policy

$$\pi_0(s) = \begin{cases} \pi_t^{(l)}(s) & \text{all of } \mathfrak{I}_t^{(1)}(s), \cdots, \mathfrak{I}_t^{(l)}(s) \text{ are true} \\ \pi^*(s) & \text{otherwise} \end{cases}$$

By lemmas 9 and 19 we have that for any $1 \leq i \leq D_{t,h}^{(l)}$, the trajectory starting from $(s_{h,i}^{(l)}, a_{h,i}^{(l)})$ using policy $\pi_t^{(l)}$ is the same with the one using policy $\pi_0$. Starting from $(s_{h_1}^*, a_{h_1}^*)$ using policy $\pi_0$, the trajectory will reach $s_{h_2}^*$, which is not on any of the trajectories starting from $(s_{h,i}^{(l)}, a_{h,i}^{(l)})$ because $\mathfrak{I}_t^{(l)}(s_{h_2}^*)$ is not true.

Using the induction hypothesis we have $a_{h_1}^* \in \mathcal{A}_t^{(l)}(s_{h_1}^*)$, so $\left\|\phi(s_{h_1}^*, a_{h_1}^*)\right\|_{\Sigma_{h, D_{t,h}^{(l)}}^{-1}} \leq 2^{-l}$. By lemma 18, we have

$$Q^{\pi_t^{(l)}}\left(s_{h_2}^*, \pi_t^{(l)}\left(s_{h_2}^*\right)\right)$$
$$\geq Q^{\pi_t^{(l)}}\left(s_{h_2}^*, \pi^*\left(s_{h_2}^*\right)\right) - 2\left(\kappa + \left(\sqrt{\lambda d}H + \sqrt{D_{t,h_1}^{(l)}}\kappa\right) \cdot 2^{-l}\right)$$
$$\geq Q^{\pi_t^{(l)}}\left(s_{h_2}^*, \pi^*\left(s_{h_2}^*\right)\right) - 2\Delta_l$$

Therefore, we conclude that $Q^{\pi_t^{(l)}}(s_{h'}^*, \pi_t^{(l)}(s_{h'}^*)) \geq Q^{\pi_t^{(l)}}(s_{h'}^*, \pi^*(s_{h'}^*)) - 2\Delta_l$ for all $h < h' \leq H$. By performance difference lemma, we have

$$Q^{\pi_t^{(l)}}(s, a) - Q^{\pi^*}(s, a)$$
$$= \sum_{h'=h+1}^{H} \left(Q^{\pi_t^{(l)}}(s_{h'}^*, \pi_t^{(l)}(s_{h'}^*)) - Q^{\pi_t^{(l)}}(s_{h'}^*, \pi^*(s_{h'}^*))\right)$$
$$\geq 2(H - h)\Delta_l$$

Furthermore,

$$Q_t^{(l)}(s, \pi^*(s))$$
$$\geq Q^{\pi_t^{(l)}}(s, \pi^*(s)) - \Delta_l \qquad\qquad \text{(lemma 16)}$$

$$\geq Q^{\pi^*}(s, \pi^*(s)) - (2(H-h)+1)\Delta_l$$

$$\geq Q^{\pi_t^{(l)}}\left(s, \pi_t^{(l)}(s)\right) - (2(H-h)+1)\Delta_l \qquad \text{(by the optimality of } \pi^*)$$

$$\geq Q_t^{(l)}\left(s, \pi_t^{(l)}(s)\right) - 2(H-h+1)\Delta_l \qquad \text{(lemma 16)}$$

so $\pi^*(s) \in \mathcal{A}_t^{(l+1)}(s)$, which concludes the induction. $\qquad\square$

**Lemma 21.** *Under $\mathfrak{E}_{high}$, for any $t \geq 1, h \in [H]$, it holds that*

$$\mathbb{E}_{t-1}\left[\sum_{i=h}^{H} r_i^{(t)}\right] \geq V^{\pi^*}\left(s_h^{(t)}\right) - 4H\sum_{i=h}^{H}\Delta_{l_i^{(t)}-1}$$

*Proof.* We use induction on $h$ from $H$ to 1.

$$\mathbb{E}_{t-1}\left[\sum_{i=h}^{H} r_i^{(t)}\right]$$

$$= \mathbb{E}_{t-1}\left[r_h^{(t)}\right] + \mathbb{E}_{t-1}\left[\sum_{i=h+1}^{H} r_i^{(t)}\right]$$

$$\geq \mathbb{E}_{t-1}\left[r_h^{(t)}\right] + V^{\pi^*}\left(s_{h+1}^{(t)}\right) - 4H\sum_{i=h+1}^{H}\Delta_{l_i^{(t)}-1} \qquad \text{(by induction)}$$

$$= Q^{\pi^*}\left(s_h^{(t)}, a_h^{(t)}\right) - 4H\sum_{i=h+1}^{H}\Delta_{l_i^{(t)}-1}$$

$$Q^{\pi^*}\left(s_h^{(t)}, a_h^{(t)}\right)$$

$$\geq Q^{\pi_t^{\left(l_h^{(t)}-1\right)}}\left(s_h^{(t)}, a_h^{(t)}\right)$$

$$\geq Q_t^{\left(l_h^{(t)}-1\right)}\left(s_h^{(t)}, a_h^{(t)}\right) - \Delta_{l_h^{(t)}-1} \qquad \text{(lemma 16)}$$

$$\geq Q_t^{\left(l_h^{(t)}-1\right)}\left(s_h^{(t)}, \pi_t^{\left(l_h^{(t)}-1\right)}\left(s_h^{(t)}\right)\right) - (2H+1)\Delta_{l_h^{(t)}-1} \qquad \text{(by the definition of } \mathcal{A}_t^{\left(l_h^{(t)}\right)})$$

$$\geq Q_t^{\left(l_h^{(t)}-1\right)}\left(s_h^{(t)}, \pi^*\left(s_h^{(t)}\right)\right) - (2H+1)\Delta_{l_h^{(t)}-1} \qquad \text{(by the definition of } \pi_t^{\left(l_h^{(t)}-1\right)})$$

$$\geq Q_t^{\pi_t^{\left(l_h^{(t)}-1\right)}}\left(s_h^{(t)}, \pi^*\left(s_h^{(t)}\right)\right) - (2H+2)\Delta_{l_h^{(t)}-1} \qquad \text{(lemma 16)}$$

$$\geq Q^{\pi^*}\left(s_h^{(t)}, \pi^*\left(s_h^{(t)}\right)\right) - 4H\Delta_{l_h^{(t)}-1} \qquad \text{(lemma 20)}$$

$$= V^{\pi^*}\left(s_h^{(t)}\right) - 4H\Delta_{l_h^{(t)}-1}$$

$\qquad\square$

**Lemma 22.** *For any $1 \leq k < \overline{L}$, it holds that*

$$\sum_{t=1}^{\infty} \mathbb{1}\left\{\exists h \in [H] \text{ s.t. } l_h^{(t)} \leq k\right\} \leq H\sum_{l=1}^{k} D_l$$

*Proof.* For any $t \geq 1$ such that $l_{h_0}^{(t)} \leq k$ for some $h_0 \in [H]$, the size of exactly one $\mathcal{D}_h^{(l)}$ will increase by 1 after the $t$-the episode for some $l \leq k$ and $h \in [H]$. Then by lemma 8 we have for any $T \geq 1$,

$$\sum_{t=1}^{T} \mathbb{1}\left\{\exists h \in [H] \text{ s.t. } l_h^{(t)} \leq k\right\} \leq \sum_{l=1}^{k}\sum_{h=1}^{H} D_{T,h}^{(l)} \leq H\sum_{l=1}^{k} D_l$$

$\square$

**Lemma 23.** *For any $h \in [H]$, it holds that $\sum_{t=1}^{T} 2^{-l_h^{(t)}} = \widetilde{\mathcal{O}}\left(\sqrt{dHT} + \frac{T\kappa}{\sqrt{H}}\right)$.*

*Proof.* Let $L$ be $\left\lceil \frac{1}{2}\log_2 \frac{T}{2Hd} \right\rceil$ so that $HD_L \geq T$. By lemma 22 we have for any $1 \leq k < \overline{L}$,

$$\sum_{t=1}^{T} \mathbb{1}\left\{ l_h^{(t)} \leq k \right\} \leq \min\left\{ T, H\sum_{l=1}^{k} D_l \right\} \leq H\min\left\{ D_L, \sum_{l=1}^{k} D_l \right\} \leq H\sum_{l=1}^{\min\{k,L\}} D_l$$

Therefore,

$$\sum_{t=1}^{T} 2^{-l_h^{(t)}} \leq \sum_{t=1}^{T} \left( 2^{-\overline{L}} + \sum_{k=1}^{\overline{L}-1} \left(2^{-k} - 2^{-(k+1)}\right) \mathbb{1}\left\{ l_h^{(t)} \leq k \right\} \right)$$

$$\leq \frac{T\kappa}{\sqrt{H}} + H\sum_{l=1}^{L} D_l \cdot 2^{-l}$$

$$= \widetilde{\mathcal{O}}\left( \frac{T\kappa}{\sqrt{H}} + \sqrt{dHT} \right)$$

$\square$

*Proof of Theorem 2.* By lemma 13, $\mathfrak{E}_{\text{high}}$ holds with probability at least $1 - \delta$, so we have

$$\text{Reg}(T) \leq 4H\sum_{h=1}^{H}\sum_{t=1}^{T} \Delta_{l_h^{(t)}-1} \qquad\qquad \text{(lemma 21)}$$

$$\leq H\sum_{h=1}^{H}\sum_{t=1}^{T} \widetilde{\mathcal{O}}\left( \sqrt{dH}\left(2^{-l_h^{(t)}} + \frac{\kappa}{\sqrt{H}}\right) \right)$$

$$\leq H\sum_{h=1}^{H} \widetilde{\mathcal{O}}\left( \sqrt{dH}\left(\sqrt{dHT} + \frac{T\kappa}{\sqrt{H}}\right) \right) \qquad\qquad \text{(lemma 23)}$$

$$= \widetilde{\mathcal{O}}\left( \sqrt{d^2H^6T} + \sqrt{d}H^2T\kappa \right)$$

$\square$

## B.6 UNIFORM-PAC

**Theorem 3.** *Let $N(\varepsilon)$ be the number of episodes whose suboptimality gap is greater than $\varepsilon$. There exists a constant $C$ such that with probability at least $1 - \delta$, it holds for all $\varepsilon > 0$ that*

$$N\left( \varepsilon + C\sqrt{d}H^2\kappa\log\frac{H}{\kappa} \right) = \widetilde{\mathcal{O}}\left( \frac{d^2H^6}{\varepsilon^2} \right)$$

*Proof.* Let $k$ be $\min\left\{ \overline{L} - 1, \log_2\left(\frac{4H^2}{\varepsilon}\sqrt{dH}\right) + \Theta\left(\log\log\frac{H}{\delta}\right) \right\}$ so that there exists a constant $C$ such that $\Delta_l \leq \frac{\varepsilon}{4H^2} + \frac{C}{4}\sqrt{d}\kappa\log\frac{H}{\kappa}$ for all $k \leq l < \overline{L}$. With probability at least $1 - \delta$, $\mathfrak{E}_{\text{high}}$ is true, so we have

$$N\left( \varepsilon + C\sqrt{d}H^2\kappa\log\frac{H}{\kappa} \right)$$

$$\leq \sum_{t=1}^{\infty} \mathbb{1}\left\{ 4H\sum_{h=1}^{H} \Delta_{l_h^{(t)}-1} > \varepsilon + C\sqrt{d}H^2\kappa\log\frac{H}{\kappa} \right\} \qquad\qquad \text{(lemma 21)}$$

$$\leq \sum_{t=1}^{\infty} \mathbb{1}\left\{ \exists h \in [H] \text{ s.t. } \Delta_{l_h^{(t)}-1} > \frac{\varepsilon}{4H^2} + \frac{C}{4}\sqrt{d}\kappa\log\frac{H}{\kappa} \right\}$$

$$\leq \sum_{t=1}^{\infty} \mathbb{1}\left\{\exists h \in [H] \text{ s.t. } l_h^{(t)} \leq k\right\} \qquad \text{(by the definition of } k)$$

$$\leq H \sum_{l=1}^{k} D_l \qquad \text{(lemma 22)}$$

$$= \tilde{\mathcal{O}}\left(\frac{d^2 H^6}{\varepsilon^2}\right)$$

$$\square$$

## C  FUNCTOIN CLASS WITH BOUNDED ELUDER DIMENSION

We extend the $Q$-function into a more general function class. Let $\mathcal{F} = \{f_\rho : \mathcal{S} \times \mathcal{A} \to [0, H] | \rho \in \Theta\}$ be a real-valued function class.

**Assumption 4.** *For any policy $\pi$ there exist $\theta_1^\pi, \cdots, \theta_H^\pi \in \Theta$ such that $f_{\theta_h^\pi}(s, a) = Q^\pi(s, a)$ for all $s \in \mathcal{S}_h, a \in \mathcal{A}$.*

**Definition 1.** *A state-action pair $(s, a) \in \mathcal{S} \times \mathcal{A}$ is $\varepsilon$-dependent on $\{(s_i, a_i)\}_{i=1}^n \subseteq \mathcal{S} \times \mathcal{A}$ w.r.t. $\mathcal{F}$ if any pair of functions $f_1, f_2 \in \mathcal{F}$ satisfying $\sum_{i=1}^n (f_1(s_i, a_i) - f_2(s_i, a_i))^2 \leq \varepsilon^2$ also satisfies $|f_1(s, a) - f_2(s, a)| \leq \varepsilon$. Further, $(s, a)$ is $\varepsilon$-independent of $\{(s_i, a_i)\}_{i=1}^n$ w.r.t. $\mathcal{F}$ if $(s, a)$ is not $\varepsilon$-dependent on $\{(s_i, a_i)\}_{i=1}^n$.*

**Definition 2.** *The $\varepsilon$-eluder dimension $\dim_E(F, \varepsilon)$ is the length $d$ of the longest sequence of elements in $\mathcal{S} \times \mathcal{A}$ such that, for some $\varepsilon' \geq \varepsilon$, every element is $\varepsilon'$-independent of its predecessors.*

Define the width of a subset $\tilde{\mathcal{F}} \subseteq \mathcal{F}$ at $(s, a) \in \mathcal{S} \times \mathcal{A}$ as

$$w_{\tilde{\mathcal{F}}}(s, a) = \sup_{\underline{f}, \overline{f} \in \tilde{\mathcal{F}}} \left(\overline{f}(s, a) - \underline{f}(s, a)\right).$$

For a dataset $\mathcal{D} = \{(s_i, a_i)\}_{i=1}^n$ and $\tilde{\mathcal{F}} \subseteq \mathcal{F}$, define

$\text{Cover}(\mathcal{D}, \tilde{\mathcal{F}}, \varepsilon) = \left\{(s, a) \in \mathcal{S} \times \mathcal{A} : (s, a) \text{ is } \varepsilon\text{-dependent on } \{(s_i, a_i)\}_{i=1}^n \text{ w.r.t. } \mathcal{F}, \text{ and } w_{\tilde{\mathcal{F}}}(s, a) \leq \varepsilon\right\}.$

Let $\beta_t^*(\mathcal{F}, \delta, \alpha)$ be

$$8H \ln \frac{N(\mathcal{F}, \alpha, \|\cdot\|_\infty)}{\delta} + 2\alpha t \left(8H + \sqrt{8H \ln \frac{4t^2}{\delta}}\right).$$

where $N(\mathcal{F}, \alpha, \|\cdot\|_\infty)$ denotes the $\alpha$-covering number of $\mathcal{F}$ in the sup-norm $\|\cdot\|_\infty$.

Define least squares estimates and confidence sets as

$$\hat{f}_{h,k}^{(l)} = \arg\min_{f \in \mathcal{F}} \sum_{i=1}^{k} \left(f(s_{h,i}^{(l)}, a_{h,i}^{(l)}) - q_{h,i}^{(l)}\right)^2$$

$$\mathcal{F}_{h,k}^{(l)} = \left\{f \in \mathcal{F} : \sum_{i=1}^{k} \left(f(s_{h,i}^{(l)}, a_{h,i}^{(l)}) - \hat{f}_{h,k}^{(l)}(s_{h,i}^{(l)}, a_{h,i}^{(l)})\right)^2 \leq \beta_k^*\left(\mathcal{F}, \frac{\delta}{4Hl^2}, \frac{1}{T^2}\right)\right\}$$

We use the same procedure as Algorithm 2 while for every $h \in [H], s \in \mathcal{S}_h, a \in \mathcal{A}$, re-define

$$\mathfrak{I}_t^{(l)}(s) = \mathbb{I}\left\{\exists 1 \leq k \leq D_{t,h}^{(l)} \text{ s.t. } (s, a) \in \text{Cover}\left(\left\{(s_{h,i}^{(l)}, a_{h,i}^{(l)})\right\}_{i=1}^{k}, \mathcal{F}_{h,k}^{(l)}, 2^{-l}\right) \text{ for all } a \in \mathcal{A}_t^{(l)}(s)\right\}$$

$$Q_t^{(l)}(s, a) = \hat{f}_{h, k_t^{(l)}(s)}(s, a)$$

and re-define constants

$$D_l = \Theta\left(2^{2l} H \dim_E\left(\mathcal{F}, T^{-1}\right)\left(\log \frac{HT}{\delta} + \log N(\mathcal{F}, T^{-2}, \|\cdot\|_\infty)\right)\right)$$

$$\Delta_l = 2^{-l}$$

$$\overline{L} = \lfloor \log_2 T \rfloor + 1$$

**Lemma 24.** *For any $t \geq 1, 1 \leq l < \overline{L}, h \in [H]$, it holds that $D_{t,h}^{(l)} \leq D_l$.*

*Proof.* Denote $D_{t,h}^{(l)}$ by $D$.

$$
\begin{aligned}
D &\leq \sum_{i=1}^{D} \mathbb{1}\left\{ w_{\mathcal{F}_{h,i}^{(l)}}\left(s_{h,i}^{(l)}, a_{h,i}^{(l)}\right) > 2^{-l} \right\} + \dim_E\left(\mathcal{F}, 2^{-l}\right) \\
&\leq \left(1 + \frac{4\beta_D^*\left(\mathcal{F}, \frac{\delta}{4Hl^2}, \frac{1}{T^2}\right)}{2^{-2l}}\right) \dim_E\left(\mathcal{F}, 2^{-l}\right) + \dim_E\left(\mathcal{F}, 2^{-l}\right) \\
&\qquad\qquad\qquad\qquad\text{(Proposition 3 from Russo \& Van Roy (2013))} \\
&\leq \mathcal{O}\left(2^{2l}\left(H \log \frac{Hl \cdot N\left(\mathcal{F}, T^{-2}, \|\cdot\|_\infty\right)}{\delta} + \frac{H}{T}\sqrt{\log \frac{HT}{\delta}}\right)\right) \dim_E\left(\mathcal{F}, T^{-1}\right) \quad (l < \overline{L}) \\
&\leq D_l
\end{aligned}
$$

$\square$

**Lemma 25.** *Define event $\mathfrak{E}_{high}$ as*

$$
\left\{ \forall l \geq 1, h \in [H], k \geq 1, f_{\theta_h^{\pi_t^{(l)}}} \in \mathcal{F}_{h,k}^{(l)} \right\}
$$

*Then $\mathbb{P}[\mathfrak{E}_{high}] \geq 1 - \delta$.*

*Proof.* For every $l \geq 1, h \in [H]$, by lemma 11 and Proposition 2 from Russo & Van Roy (2013), with probability at least $1 - \frac{\delta}{2Hl^2}$, we have

$$
f_{\theta_h^{\pi_t^{(l)}}} \in \mathcal{F}_{h,k}^{(l)}
$$

for all $k \geq 1$. A union bound over $l$ and $h$ concludes the proof. $\square$

**Lemma 26.** *Under $\mathfrak{E}_{high}$, given $t \geq 1, l \geq 1, h \in [H], s \in \mathcal{S}_h$, if $\mathfrak{I}_t^{(l)}(s)$ is true, then for any $a \in \mathcal{A}_t^{(l)}(s)$,*

$$
\left| Q_t^{(l)}(s,a) - Q^{\pi_t^{(l)}}(s,a) \right| \leq 2^{-l}
$$

*Proof.* Denote $k_t^{(l)}(s,a)$ by $k$. Since $\mathfrak{I}_t^{(l)}(s)$ is true, $w_{\mathcal{F}_{h,k}^{(l)}}(s,a) \leq 2^{-l}$ for all $a \in \mathcal{A}_t^{(l)}(s)$, so we have

$$
\begin{aligned}
&\left| Q_t^{(l)}(s,a) - Q^{\pi_t^{(l)}}(s,a) \right| \\
&= \left| \hat{f}_{h,k}^{(l)}(s,a) - f_{\theta_h^{\pi_t^{(l)}}}(s,a) \right| \\
&\leq w_{\mathcal{F}_{h,k}^{(l)}}(s,a) \leq 2^{-l}
\end{aligned}
$$

$\square$

**Lemma 27.** *For $\varepsilon > 0, \beta > 0, 1 \leq h_1 < h_2 \leq H$, given a policy $\pi_0$, and $n+1$ trajectories*

$$
\left\{ \left( s_{h_1}^{(i)}, a_{h_1}^{(i)}, \cdots, s_{h_2}^{(i)}, a_{h_2}^{(i)} \right) \right\}_{i=0}^{n}
$$

*such that $\pi_0(s_h^{(i)}) = a_h^{(i)}$ for all $0 \leq i \leq n, h_1 + 1 \leq h \leq h_2$, if $(s_{h_1}^{(0)}, a_{h_1}^{(0)})$ is $\varepsilon$-dependent on $\left\{ (s_{h_1}^{(i)}, a_{h_1}^{(i)}) \right\}_{i=1}^{n}$, and $s_{h_2}^{(0)} \neq s_{h_2}^{(i)}$ for all $1 \leq i \leq n$, then for any policy $\pi$ and any $a_1, a_2 \in \mathcal{A}$, it holds that*

$$
\left| Q^\pi\left(s_{h_2}^{(0)}, a_1\right) - Q^\pi\left(s_{h_2}^{(0)}, a_2\right) \right| \leq 2\varepsilon
$$

*Proof.* Fix a policy $\pi$. Define policy $\overline{\pi}$ and $\overline{\pi}_a$ for $a \in \mathcal{A}$ as follows, where $\text{stage}(s) = h$ if $s \in \mathcal{S}_h$,

$$\overline{\pi}(s) = \begin{cases} \pi_0(s) & \text{stage}(s) \leq h_2 \\ \pi(s) & \text{stage}(s) > h_2 \end{cases}$$

$$\overline{\pi}_a(s) = \begin{cases} a & s = s_{h_2}^{(0)} \\ \overline{\pi}(s) & s \neq s_{h_2}^{(0)} \end{cases}$$

Since $s_{h_2}^{(0)} \neq s_{h_2}^{(i)}$ for all $1 \leq i \leq n$, we have $Q^{\overline{\pi}}(s_{h_1}^{(i)}, a_{h_1}^{(i)}) = Q^{\overline{\pi}_a}(s_{h_1}^{(i)}, a_{h_1}^{(i)})$. Let $R$ be

$$\mathbb{E}\left[ r_{h_1} + \cdots + r_{h_2-1} \,\Big|\, s_{h_1} = s_{h_1}^{(0)}, a_{h_1} = a_{h_1}^{(0)}, \pi_0 \right]$$

Then $Q^{\overline{\pi}_a}(s_{h_1}^{(0)}, a_{h_1}^{(0)}) = R + Q^{\pi}(s_{h_2}^{(0)}, a)$ for all $a \in \mathcal{A}$.

Since $\sum_{i=1}^n \left( f_{\theta_{h_1}^{\overline{\pi}}}(s_{h_1}^{(i)}, a_{h_1}^{(i)}) - f_{\theta_{h_1}^{\overline{\pi}_a}}(s_{h_1}^{(i)}, a_{h_1}^{(i)}) \right)^2 = 0 \leq \varepsilon^2$, we have

$$\left| f_{\theta_{h_1}^{\overline{\pi}}}\left( s_{h_1}^{(0)}, a_{h_1}^{(0)} \right) - f_{\theta_{h_1}^{\overline{\pi}_a}}\left( s_{h_1}^{(0)}, a_{h_1}^{(0)} \right) \right| \leq \varepsilon$$

by the definition of $\varepsilon$-dependence. Let $q_0$ be $Q^{\overline{\pi}}(s_{h_1}^{(0)}, a_{h_1}^{(0)}) - R$. Then we have for any $a \in \mathcal{A}$,

$$\begin{aligned}
& \left| Q^{\pi}\left( s_{h_2}^{(0)}, a \right) - q_0 \right| \\
&= \left| Q^{\pi}\left( s_{h_2}^{(0)}, a \right) + R - Q^{\overline{\pi}}\left( s_{h_1}^{(0)}, a_{h_1}^{(0)} \right) \right| \\
&= \left| Q^{\overline{\pi}_a}\left( s_{h_1}^{(0)}, a_{h_1}^{(0)} \right) - Q^{\overline{\pi}}\left( s_{h_1}^{(0)}, a_{h_1}^{(0)} \right) \right| \\
&= \left| f_{\theta_{h_1}^{\overline{\pi}_a}}\left( s_{h_1}^{(0)}, a_{h_1}^{(0)} \right) - f_{\theta_{h_1}^{\overline{\pi}}}\left( s_{h_1}^{(0)}, a_{h_1}^{(0)} \right) \right| \\
&\leq \varepsilon
\end{aligned}$$

which gives the desired result. $\qquad\square$

By lemmas 26 and 27, lemmas 20 and 21 still hold.

**Lemma 28.** *For any $h \in [H]$, it holds that*

$$\sum_{t=1}^T 2^{-l_h^{(t)}} = \mathcal{O}\left( \sqrt{H^2 T \dim_E (\mathcal{F}, T^{-1}) \left( \log \frac{HT}{\delta} + \log N(\mathcal{F}, T^{-2}, \|\cdot\|_\infty) \right)} \right)$$

*Proof.* Let $L$ be

$$\frac{1}{2} \log_2 \frac{T}{H^2 \dim_E (\mathcal{F}, T^{-1}) \left( \log \frac{HT}{\delta} + \log N(\mathcal{F}, T^{-2}, \|\cdot\|_\infty) \right)} + \Theta(1)$$

so that $HD_L \geq T$. By lemma 22 we have for any $1 \leq k < \overline{L}$,

$$\sum_{t=1}^T \mathbb{I}\left[ l_h^{(t)} \leq k \right] \leq \min\left\{ T, H \sum_{l=1}^k D_l \right\} \leq H \min\left\{ D_L, \sum_{l=1}^k D_l \right\} \leq H \sum_{l=1}^{\min\{k,L\}} D_l$$

Therefore,

$$\begin{aligned}
\sum_{t=1}^T 2^{-l_h^{(t)}} &\leq \sum_{t=1}^T \left( 2^{-\overline{L}} + \sum_{k=1}^{\overline{L}-1} \left( 2^{-k} - 2^{-(k+1)} \right) \mathbb{1}\left\{ l_h^{(t)} \leq k \right\} \right) \\
&\leq \sqrt{T} + H \sum_{l=1}^L D_l \cdot 2^{-l} \\
&= \mathcal{O}\left( \sqrt{H^2 T \dim_E (\mathcal{F}, T^{-1}) \left( \log \frac{HT}{\delta} + \log N(\mathcal{F}, T^{-2}, \|\cdot\|_\infty) \right)} \right)
\end{aligned}$$

$\qquad\square$

**Theorem 4.** *With probability at least $1 - \delta$, it holds that for any $T \geq 1$,*

$$\text{Reg}(T) = \mathcal{O}\left(\sqrt{H^6 T \dim_E\left(\mathcal{F}, T^{-1}\right)\left(\log \frac{HT}{\delta} + \log N(\mathcal{F}, T^{-2}, \|\cdot\|_\infty)\right)}\right)$$

*Proof.* By lemma 25, $\mathfrak{E}_{\text{high}}$ holds with probability at least $1 - \delta$, so we have

$$\text{Reg}(T) \leq 4H \sum_{t=1}^{T}\sum_{h=1}^{H} \Delta_{l_h^{(t)}-1} \qquad\qquad \text{(lemma 20)}$$

$$= 8H \sum_{h=1}^{H}\sum_{t=1}^{T} 2^{-l_h^{(t)}}$$

$$= \mathcal{O}\left(\sqrt{H^6 T \dim_E\left(\mathcal{F}, T^{-1}\right)\left(\log \frac{HT}{\delta} + \log N(\mathcal{F}, T^{-2}, \|\cdot\|_\infty)\right)}\right) \qquad \text{(lemma 28)}$$

$\square$

