# OpenReview forum: "Frozen Policy Iteration: Computationally Efficient RL under Linear $Q^{\pi}$ Realizability for Deterministic Dynamics"
_ICLR.cc/2026/Conference — ICLR 2026 Poster_

### Official Review · Reviewer_Cp6M · 2025-10-24

**Soundness:** 4
**Presentation:** 4
**Contribution:** 4
**Rating:** 8
**Confidence:** 3

**Summary:**

This paper makes convincing progress in developing computationally and statistically efficient algorithms to learn an optimal policy in linear $Q^\pi$ realizable MDPs, here with the additional assumptions that the dynamics are deterministic.
I checked the proofs and they look correct and in retrospect simple which is a further positive point in favour of the paper.
Moreover, the "frozen" technique seems to be useful in the experiments.

**Strengths:**

The analysis is simple and elegant.

Moreover, you need only the realizability of the policies that are played by the algorithm. This is a much milder assumption compared to the Linear $Q^\pi$ realizability assumption imposed in [1,2] which crucially relies on the realizability also of policies that are never selected by the algorithm (in order to prove that the expectation of any low range function is linear).

[1] Weisz et al. Online RL in Linearly -Realizable MDPs Is as Easy as in Linear MDPs If You Learn What to Ignore

[2] Zakaria Mhammedi, Sample and Oracle Efficient Reinforcement Learning for MDPs with Linearly-Realizable Value Functions

**Weaknesses:**

1) If I understand correctly, the freezing technique requires keeping in memory the past linear weights of the Q function. Since the number of states to be frozen will eventually be of order $\epsilon^{-2}$, it seems to me that the suggested algorithm requires memory scaling as $\epsilon^{-2} d$. Is this correct? If yes, please mention it clearly in the paper.
Similarly, also deploying the learned policy does not seem very easy because it requires keeping in memory all the history of the $Q$-values weights, and a snapshot of the dataset at the different times $t$. At deployment time, when a new state action pair is visited, it does not seem very easy to check at which time step it started to be covered. Do you have any ideas how you could learn a policy parametrized by only $d$ parameters in this setting?

2) The regret analysis holds only for stochastic rewards and not adversarially chosen. It might be better to clarify this.

3) The work "Sample and Oracle Efficient Reinforcement Learning for MDPs with Linearly-Realizable Value Functions" by Zakaria Mhammedi is very relevant and should be discussed in the related works.

**Questions:**

If the memory issue I pointed out is correct, do you see any way to run the algorithm with memory that is independent of $\mathrm{poly}(\epsilon^{-1})$?

The author never says that after $D$ updates of the dataset than any state action pair is covered. I think this remark would help a lot with the proof sketch of Theorem 1. Could you present the proof stating that the statement of Lemma 6 implies Theorem 1 if every state action pair is covered, and then state that this happens necessarily after D updates of the dataset, invoking Lemma 1?

Do you have any idea about how this algorithm could be extended to the adversarial reward setting ?

---

> ### Author Response · Authors · 2025-11-19
>
> - Q1: If the memory issue I pointed out is correct, do you see any way to run the algorithm with memory that is independent of $\operatorname{poly}(\varepsilon^{-1})$?
>
>   A: The reviewer is correct that the space complexity of Algorithm 1 scales as $\widetilde{\mathcal{O}}(H\operatorname{poly}(d)/\varepsilon^{2})$. We have explicitly clarified this in the revised version. To reduce this dependence on $\varepsilon^{-1}$, one possible approach is to adopt ideas similar to those used by [1], where policy updates occur only when the empirical covariance matrix changes a significant amount. Such method would likely limit the number of effective updates to $O(Hd\log\varepsilon^{-1})$, suggesting that the memory cost might be reduced from polynomial to logarithmic in $\varepsilon^{-1}$.
>
> - Q2: The author never says that after $D$ updates of the dataset than any state action pair is covered. I think this remark would help a lot with the proof sketch of Theorem 1. Could you present the proof stating that the statement of Lemma 6 implies Theorem 1 if every state action pair is covered, and then state that this happens necessarily after D updates of the dataset, invoking Lemma 1?
>
>   A: We thank the reviewer for this helpful suggestion. We have revised the presentation of the proof of Theorem 1 to follow the reviewer’s recommendation.
>
> - Q3: Do you have any idea about how this algorithm could be extended to the adversarial reward setting?
>
>   A: We thank the reviewer for pointing this out. We have clarified in the Preliminaries that rewards are stochastic and bounded in $[0,1]$ in the revised version. Our concentration-based analysis (via self-normalized inequalities) fundamentally relies on this stochasticity assumption to establish high-probability confidence bounds. Extending the algorithm to the adversarial reward setting would require new techniques — likely based on adversarial bandit.
>
> - Q4: The work "Sample and Oracle Efficient Reinforcement Learning for MDPs with Linearly-Realizable Value Functions" by Zakaria Mhammedi is very relevant and should be discussed in the related works.
>
>   A: We have included Mhammedi's work in the revised version.
>
> [1] Gao et al., A Provably Efficient Algorithm for Linear Markov Decision Process with Low Switching Cost.

---

> > ### Comment · Reviewer_Cp6M · 2025-11-20
> >
> > thanks a lot for the changes  and for your reply!
> >
> > I reiterate that I really appreciate your submission!
> >
> > As a last comment I think you could emphasize that the assumption used in your submission is weaker than the assumption used in Weisz et al. Indeed you do not need Q value realizability of policies that the algorithm does not play.
> >
> > Best,
> > reviewer

---

> > > ### Author Response · Authors · 2025-11-20
> > >
> > > We appreciate the reviewer's positive comment and the valuable insight regarding the linear $Q^{π}$ realizability. Indeed, our result holds under the weaker version of this assumption. We will revise the manuscript to highlight this important observation in the next revision.

---

> ### Author Response · Authors · 2025-11-25
>
> In the revised version of the paper, we have removed an $H$ factor from both the PAC and regret bounds. Please refer to the updated revision and the author comment for detailed explanations of the improvement.

---

### Official Review · Reviewer_Mz6n · 2025-10-27

**Soundness:** 3
**Presentation:** 3
**Contribution:** 3
**Rating:** 6
**Confidence:** 3

**Summary:**

This paper proposes a computationally and statistically efficient reinforcement learning algorithm, called Frozen Policy Iteration (FPI), under the linear $Q^\pi$ realizability assumption.
Unlike prior works, FPI requires no simulator access and operates with stochastic initial states and rewards, assuming only deterministic transitions.
The key idea of FPI is to freeze the policy for well-explored states—once all actions at a state are sufficiently covered, the algorithm stops updating that state’s policy. This mechanism ensures that all collected data remain effectively on-policy throughout learning, enabling efficient regret minimization without the need for resampling or simulator resets.
FPI achieves a regret bound of $\tilde{O}(\sqrt{d^2 H^7 T})$, which is remarkable since most prior works under this assumption attain only PAC-type guarantees.

**Strengths:**

Overall, the paper is well-written and easy to follow.
Achieving what appears to be the first regret bound under the linear $Q^\pi$ realizability setting is an interesting result.
However, the paper seems to require clearer articulation and positioning of its key contributions.

**Weaknesses:**

1. Weisz et al. (2023) study a similar setting, except that they allow for stochastic transitions. Their algorithm attains only a PAC guarantee, not a regret bound. If their analysis were restricted to deterministic transitions, would their algorithm also achieve a regret bound comparable to FPI? A discussion of this comparison, maybe following Theorem 1, should be included in the paper (including explicit contrasts between the PAC guarantees).

2. The computational cost of the proposed algorithm is not discussed. A comparison of the computational efficiency with prior works would strengthen the paper.

3. The proposed algorithm is limited to finite state and action spaces, which diminishes the benefits of using function approximation.

4. No experimental results. Even simple empirical validations or illustrative examples would help demonstrate the practicality of the proposed approach.

**Questions:**

1. At which point in the analysis is the deterministic transition assumption essential?

2. During exploration—when a state–action pair $(s,a)$ is insufficiently covered by the dataset—do you think it would be possible to achieve better performance by selecting actions based on *optimal design* methods rather than choosing them randomly?

---

> ### Author Response · Authors · 2025-11-19
>
> - Q1: At which point in the analysis is the deterministic transition assumption essential?
>
>   A: The current analysis critically relies on the property that all state-action pairs collected in the datasets are well-explored. This property ensures that once a pair $(s,a)$ is added to the dataset, the subsequent trajectory remains within the high-confidence region, preserving the “effectively on-policy” nature of the data. We have added a detailed explanation of this point to the Discussion section in the revised version.
>
> - Q2: During exploration — when a state-action pair $(s,a)$ is insufficiently covered by the dataset — could performance be improved by selecting actions via optimal design methods instead of random choices?
>
>   A: We appreciate the suggestion. In our current analysis, the dataset size is upper bounded by the elliptical potential lemma. Incorporating optimal design methods is indeed an interesting direction, particularly for settings with large or infinite action spaces. We view this as a promising method, especially for improving computation performance in high-dimensional or continuous action domains.
>
> - Q3: Weisz et al. (2023) study a similar setting, except that they allow for stochastic transitions. Their algorithm attains only a PAC guarantee, not a regret bound. If their analysis were restricted to deterministic transitions, would their algorithm also achieve a regret bound comparable to FPI? A discussion of this comparison, maybe following Theorem 1, should be included in the paper.
>
>   A: We thank the reviewer for this comparison.  The algorithm of Weisz et al. (2023) is substantially more complex than ours and relies on a global optimism-based procedure. Naively applying their algorithm can only achieve a $T^{2/3}$-regret bound, and it's unclear how to improve such bound. Moreover, their approach is not computationally efficient. In contrast, our algorithm is computationally efficient while still achieving both regret and Uniform-PAC guarantees under deterministic dynamics. We have clarified this contrast in the revision.
>
> - Q4: The computational cost of the proposed algorithm is not discussed.
>
>   A: The running time of our algorithm is polynomial in $d$, $H$, $T$, and the size of action space. We have added a discussion of the computational complexity in the revised version.
>
> - Q5: The proposed algorithm is limited to finite state and action spaces, which diminishes the benefits of using function approximation.
>
>   A: We would like to clarify that our algorithm only requires a finite action space, while the state space can be infinite or continuous, as long as a finite-dimensional feature representation is provided. This is consistent with the standard linear-function-approximation setting.
>
> - Q6: No experimental results. Even simple empirical validations or illustrative examples would help demonstrate the practicality of the proposed approach.
>
>   A: We clarify that we do include experiments on Algorithm 1 (the PAC version) in Section 6. These experiments demonstrate the practical effectiveness of the proposed method.

---

> ### Author Response · Authors · 2025-11-25
>
> In the revised version of the paper, we have removed an $H$ factor from both the PAC and regret bounds. Please refer to the updated revision and the author comment for detailed explanations of the improvement.

---

> > ### Comment · Reviewer_Mz6n · 2025-11-27
> >
> > Thank you for the response and for conducting additional experiments despite the limited time. However, I still wonder whether the experiments are sufficient; the authors should consider including other baseline algorithms. In the current version, I remain unconvinced about the practicality of the proposed approach. That said, I am not requesting additional experiments during this discussion period. As it stands, I believe the theoretical contribution is strong enough, and I will therefore maintain my positive score.

---

### Official Review · Reviewer_TPzh · 2025-11-01

**Soundness:** 2
**Presentation:** 2
**Contribution:** 2
**Rating:** 4
**Confidence:** 3

**Summary:**

In online reinforcement learning with deterministic transitions, the authors propose frozen policy iteration under linear $Q^\pi$ realizability. The method admits only high-confidence part of each trajectory into the dataset and freezes the policy at sufficiently explored states and learn without restarting simulator or resampling. They prove high-probability regret bounds, uniform-PAC guarantees, and extensions based on the eluder dimension, and demonstrate the effectiveness of freezing on small-scale control tasks.

**Strengths:**

1. They present a computationally efficient algorithm that operates without simulator restart or resampling, and provide theoretical guarantees via high probability regret bounds, Uniform-PAC results, and extension to function classes based on the eluder dimension.
2. They conduct ablation studies on the effect of freezing across two RL environments and report detailed implementation choices to aid reproducibility.

**Weaknesses:**

1. The theory relies heavily on linear $Q^\pi$ realizability (Assumption 1) and deterministic transitions (Assumption 3).
2. The regret bound and Uniform-PAC bound exhibit significant dependence on the horizon $H$.
3. PAC guarantees may become loose over practically interesting $\varepsilon$ ranges when $\kappa$ is not small.
4. The experiments are limited to verifying Algorithm 1 and ablation studies on freezing, while Algorithm 2—one of the core components of the proposed theory—was neither implemented nor evaluated.

**Questions:**

1. I am curious how the strong dependency on $H$ arises in the regret or Uniform-PAC bounds, and whether there is scope for mitigation.
2. In the Algorithm 1 experiments, is it possible to compare with additional baseline algorithms beyond the freezing ablation? Could you also provide experimental validation for Algorithm 2?

---

> ### Author Response · Authors · 2025-11-19
>
> - Q1: In the Algorithm 1 experiments, is it possible to compare with additional baseline algorithms beyond the freezing ablation? Could you also provide experimental validation for Algorithm 2?
>
>   A: As discussed in the paper (Section 2), prior works under the same linear $Q^{\pi}$ realizability assumption either rely on computationally intractable optimization or require access to a simulator. These methods are therefore not implementable in the standard online RL setting that we study, where resampling or simulator access are not allowed. Existing efficient algorithms rely on the linear Bellman completeness assumption, which is different from the linear $Q^\pi$ realizability assumption we study. As a result, those methods are not applicable to our setting.
>
>   Algorithm 1 and Algorithm 2 share the same core mechanism of policy freezing. Algorithm 2 mainly adds multiple accuracy levels for regret minimization, but this introduces substantial programming complexity, making a faithful implementation non-trivial. Since the added components primarily serve theoretical purposes rather than introducing new empirical phenomena, we believe additional experiments are unnecessary to support the theoretical claims of the paper.
>
> - Q2: How does the strong dependency on $H$ arise in the regret or Uniform-PAC bounds? Whether there is scope for mitigation?
>
>   A: The relatively high polynomial dependence on $H$ in the regret and Uniform-PAC bounds primarily arises from the need for exploration under multiple accuracy level constraints. Ensuring that the optimal action of each state is preserved across all accuracy levels introduces additional $H$ factors compared to the PAC analysis. In the revised version, we have tightened the Uniform-PAC bound from $\widetilde{\mathcal{O}}(\frac{d^2H^8}{\varepsilon^2})$ to $\widetilde{\mathcal{O}}(\frac{d^2H^7}{\varepsilon^2})$. Further improvements of dependence on $H$ will be an important focus of future work.
>
> - Q3: PAC guarantees may become loose over practically interesting $\varepsilon$ ranges when $\kappa$ is not small.
>
>   A: As shown by [1], even in the simpler bandit setting the suboptimality gap can only reach $\mathcal{O}(\sqrt{d}\kappa)$. Our PAC bound reduces to the same dependence when $H = 1$. Since the finite-horizon MDP setting is strictly more challenging than the bandit case, obtaining guarantees that qualitatively improve on this $\kappa$-dependence is non-trivial.
>
> [1] Lattimore et al., Learning with Good Feature Representations in Bandits and in RL with a Generative Model.

---

> ### Author Response · Authors · 2025-11-25
>
> In the revised version of the paper, we have removed an $H$ factor from both the PAC and regret bounds. Please refer to the updated revision and the author comment for detailed explanations of the improvement.

---

### Official Review · Reviewer_Jqpd · 2025-11-05

**Soundness:** 3
**Presentation:** 2
**Contribution:** 3
**Rating:** 6
**Confidence:** 4

**Summary:**

The submission employs algorithmic peeling for the exploration uncertainty to achieve the first sublinear regret with tractable computational complexity for learning in MDPs with linear $Q^\pi$ realizability under deterministic transition and step-wise disjoint state spaces. Their analysis bypass the need of simulations of "rollout reset" in the online setting and naturally compatible with the Uniform-PAC analysis given the multi-level regression structure is similar to that in He et al. 2021 (Uniform-pac bounds for reinforcement learning with linear function approximation)

**Strengths:**

- The delicate exploitation of the loopless deterministic "tree" structure via peeling and the design of $k_t(s)$ is novel
- The multi-level uncertainty slicing in Algorithm 2 aligns well with the intuition and is easy to follow

**Weaknesses:**

- On the disjointness assumption of $\mathcal{S}$: It seems that the regret analysis heavily rely on the disjointness assumption of the state space, i.e., $\mathcal{S}\_{h_1} \cap \mathcal{S}\_{h_2} = \emptyset$ when $h_1 \neq h_2$, effectively enforcing the process to be a "tree", e.g., in **the proof of Lemma 18**, etc. This is an unusually strong assumption because such an assumption is often considered acceptable only in adversarial MDPs.

### Minor weaknesses

- According to the definition of $\bar{L}$, the time complexity of Algorithm 2 appears to be finite only if $\kappa > 0$, which is counter-intuitive. This might be only a matter of presentation but the reviewer encourages the authors to elaborate it in the final version.
- Line 052: To the best of the reviewer's knowledge, MDPs with linear bellman completeness are only known to be tractable under deterministic transitions [1] or constant many actions [2]
- On Lemma 12: The authors did not mention (in the main text) their assumption on reward noise in Preliminaries or Theorem 2.
- Reindexing $\mathcal{D}_{t,h}$ is acceptable but its notation should be detailed before Section 4.1

References

[1] Wu, Runzhe, et al. "Computationally efficient rl under linear bellman completeness for deterministic dynamics." arXiv preprint arXiv:2406.11810 (2024).

[2] Golowich, Noah, and Ankur Moitra. "Linear bellman completeness suffices for efficient online reinforcement learning with few actions." The Thirty Seventh Annual Conference on Learning Theory. PMLR, 2024.

**Questions:**

- Although it is acceptable for a theory paper to not implement Algorithm 2, the fact that Algorithm 2 requires $\kappa$ as an **input**, e.g., in the computations of $\bar{L}$ and $\Delta_l$'s, is indeed a concern since it is unclear how to estimate $\kappa$ a priori. Do the authors plan to justify or make some educated guess on the mitigation of this fact?
- Line 269: the term "near-optimal" might be misleading, I guess the authors mean low-$\mathcal{D}_{t, h}$-uncertainty for $s_h^{(t)}$?
- On the $\sqrt{d}H$ in Assumption 2: let us say $\kappa$ is very close to $0$, then what would be the upper bound of $Q^\pi$ in Assumption 1, i.e., the upper bound of $Q^\pi$ in Assumption 1 then becomes $\sqrt{d}H$?

---

> ### Author Response · Authors · 2025-11-19
>
> - Q1: Although it is acceptable for a theory paper to not implement Algorithm 2, the fact that Algorithm 2 requires $\kappa$ as an input, e.g., in the computations of $\overline{L}$ and $\Delta_l$'s, is indeed a concern since it is unclear how to estimate $\kappa$ a priori. Do the authors plan to justify or make some educated guess on the mitigation of this fact?
>
>   A: We acknowledge the reviewer’s concern. A practical approach is to run Algorithm 2 using a range of candidate values for $\kappa$ and select the value that yields the best empirical performance.
>
> - Q2: Line 269 — the term “near-optimal” might be misleading, I guess the authors mean low-$D_{t,h}$-uncertainty for $s_h^{(t)}$?
>
>   A: The reviewer is correct. The intended meaning is low-$D_{t,h}$-uncertainty rather than “near-optimal.” We have revised the text to use clearer terminology and avoid confusion.
>
> - Q3: On the $\sqrt{d}H$ in Assumption 2: let us say $\kappa$ is very close to $0$, then what would be the upper bound of $Q^{\pi}$ in Assumption 1, i.e., the upper bound of $Q^{\pi}$ in Assumption 1 then becomes $\sqrt{d}H$? On Lemma 12 — The authors did not mention (in the main text) their assumption on reward noise in Preliminaries or Theorem 2.
>
>   A: We have clarified in the Preliminaries that rewards are bounded within $[0,1]$, and hence the noise term is also bounded and $Q^\pi$ naturally lies in $[0,H]$.
>
> - Q4: On the disjointness assumption of $\mathcal{S}$.
>
>   A: The disjointness assumption of the state space across stages is made without loss of generality. In episodic finite-horizon MDPs, one can always view the same state appearing at different stages as distinct elements, i.e., treat $(s,h)$ as a unique stage-dependent state. This transformation makes the state sets disjoint while preserving the transition and reward structure.
>
> - Q5: According to the definition of $\overline{L}$, the time complexity of Algorithm 2 appears to be finite only if $\kappa>0$, which is counter-intuitive. This might be only a matter of presentation but the reviewer encourages the authors to elaborate it in the final version.
>
>   A: We have added a clarification in the revision: although the definition of $\overline{L}$ involves $\kappa$, it only affects the theoretical range of accuracy levels and does not influence the actual computational time of Algorithm 2. The running time remains polynomial in $d$, $H$, $T$, and the size of action space, regardless of $\kappa$.
>
> - Q6: Line 052 — To the best of the reviewer's knowledge, MDPs with linear bellman completeness are only known to be tractable under deterministic transitions or constant many actions.
>
>   A: At that point in the text, we were referring to statistically efficient algorithms (i.e., those with polynomial sample complexity), not computationally efficient ones.

---

> ### Comment · Reviewer_Jqpd · 2025-11-21
>
> Thank you for the detailed response. I keep my positive evaluation unchanged and vote for the acceptance of this submission.
>
> > As a minor suggestion: even if the authors were referring to statistically efficient algorithms under linear Bellman completeness in Line 052, as far as I know, [Theorem 5.10, https://arxiv.org/pdf/2406.11640] is by far (for non-deterministic transitions) the best sample complexity; which is exponential in $|\mathcal{A}|$.

---

> > ### Author Response · Authors · 2025-11-23
> >
> > Thanks for your suggestion! Under linear Bellman completeness, if we allow computationally inefficient algorithms, polynomial sample complexity (polynomial in $H$, $d$, $|A|$ and $1 / \epsilon$) is actually known in the literature. See e.g. [1] which was also mentioned in the abstract of the paper by Golowich and Moitra.
> >
> > [1] Andrea Zanette, Alessandro Lazaric, Mykel Kochenderfer, and Emma Brunskill. Learning near optimal policies with low inherent Bellman error. ICML 2020

---

> > > ### Comment · Reviewer_Jqpd · 2025-11-23
> > >
> > > Thank you for your response. I have got the point. I keep my positive evaluation unchanged and vote for the acceptance of this submission.

---

> ### Author Response · Authors · 2025-11-25
>
> In the revised version of the paper, we have removed an $H$ factor from both the PAC and regret bounds. Please refer to the updated revision and the author comment for detailed explanations of the improvement.

---

### Author Response · Authors · 2025-11-19

We thank the reviewers for recognizing our contribution and for their thoughtful and detailed feedback. We have updated a revised version of the paper, with changes highlighted in red. In particular, we have added a detailed discussion of the computational costs of our algorithms and a new section outlining key discussions and open problems.

---

### Author Response · Authors · 2025-11-25

We have updated a revised version, with changes marked in blue. We have reduced the $H$-dependence in both the PAC and regret bounds to $\widetilde{\mathcal{O}}(\frac{d^2H^4}{\overline{\varepsilon}^2})$ and $\widetilde{\mathcal{O}}(\sqrt{d^2H^6T})$, respectively. The improvement relies on the determinism of the MDP transitions. Because both $\pi_t$ and the MDP transitions are deterministic, the trajectory at round $t$ is determined given $\pi_t$. Consequently, the noise term $\xi_t$ is the sum of at most $H$ independent $1$-subGaussians, and is therefore $\sqrt{H}$-subGaussian.

---

### Meta-Review · Area_Chair_L2J3 · 2026-01-04

**Summary:**

This paper tackles the online RL setting under linear $Q^\pi$ realizability with deterministic transitions (allowing stochastic initial states and rewards), where prior policy-iteration style approaches often rely on simulator access or the ability to revisit the same state repeatedly. Their proposed method, Frozen Policy Iteration (FPI), is built around a sensible idea: only trust high-confidence parts of trajectories and freeze the policy at states once the action set there is sufficiently covered, so the algorithm can keep learning without drifting into uncontrolled off-policy data. The paper provides regret guarantees, a Uniform-PAC variant, and an extension via eluder dimension, plus a small proof-of-concept implementation/ablation supporting the role of freezing.

The paper provides a good theoretical contribution, providing computationally efficient online RL under linear $Q^\pi$
 realizability, avoiding simulator resets and repeated rollouts from the same state.
Several reviewers found the analysis simple/elegant, and the freezing mechanism is easy to understand as a way of keeping the data effectively on-policy. The paper also does a decent job connecting the regret result to a structured PAC warm-up, and the additional Uniform-PAC / eluder-dimension extensions.

**Reviewer Concerns:**

The recurring reservations are about scope and practicality, rather than correctness.
The setting relies heavily on deterministic dynamics and the realizability assumption; reviewers correctly point out that this narrows applicability, and extending beyond determinism is not straightforward.
The regret/Uniform-PAC bounds still have a large polynomial dependence on $H$, which makes the theoretical guarantees less satisfying for long-horizon problems.
A reviewer pointed out that the paper initially underplayed practical details: computational and memory costs (especially due to freezing). The updated version includes the discussion and comment about the time and space complexity.
Empirically, the experiments are rather weak. They help sanity-check the freezing idea (and the ablation is useful), but they don’t yet make a strong case for practicality beyond toy control tasks, and there are limited comparisons, although it is also true that truly comparable baselines under the same access model are not easy to implement.

The authors engaged constructively and addressed several of the above in the revision.

**Reviewer Scores:**

Overall, I think the paper is a reasonable accept. The assumptions may be narrow, but within that stated regime, the result is nontrivial and, importantly, the algorithmic idea is interesting enough that it may be useful beyond this exact setting. The remaining weaknesses do exist, but they don’t outweigh the core theoretical step forward.

---

### Decision · Program_Chairs · 2026-01-26

Accept (Poster)